# Growing season CH$_4$ and N$_2$O fluxes from a sub-arctic landscape in northern Finland; from chamber to landscape scale

Dinsmore, K. J.[1], Drewer, J.[1], Levy, P.E.[1], George, C.[2], Lohila, A.[3], Aurela, M.[3], Skiba, U.M[1]

[1]Centre for Ecology and Hydrology, Penicuik, EH26 0QB, UK
[2]Centre for Ecology and Hydrology, Wallingford OX10 8BB, UK
[3]Finnish Meteorological Institute, Atmospheric Composition Research, FI-00101 Helsinki, Finland

*Correspondence to*: Ute Skiba (ums@ceh.ac.uk); Kerry J Dinsmore (kjdi@ceh.ac.uk)

**Abstract.** Subarctic and boreal emissions of CH$_4$ are important contributors to the atmospheric greenhouse gas (GHG) balance and subsequently the global radiative forcing. Whilst N$_2$O emissions may be lower, the much greater radiative forcing they produce justifies their inclusion in GHG studies. In addition to the quantification of flux magnitude, it is essential that we understand the drivers of emissions to be able to accurately predict climate-driven changes and potential feedback mechanisms. Hence this study aims to increase our understanding of what drives fluxes of CH$_4$ and N$_2$O in a subarctic forest/wetland landscape during peak summer conditions and into the shoulder season, exploring both spatial and temporal variability, and uses satellite derived spectral data to extrapolate from chamber scale fluxes to a 2 x 2 km landscape area.

From static chamber measurements made during summer and autumn campaigns in 2012 in the Sodankylä region of Northern Finland, we concluded that wetlands represent a significant source of CH$_4$ (3.35 ± 0.44 mg C m$^{-2}$ hr$^{-1}$ during summer campaign and 0.62 ± 0.09 mg C m$^{-2}$ hr$^{-1}$ during autumn campaign), whilst the surrounding forests represent a small sink (-0.06 ± <0.01 mg C m$^{-2}$ hr$^{-1}$ during the summer campaign and -0.03 ± <0.01 mg C m$^{-2}$ hr$^{-1}$ during the autumn campaign). N$_2$O fluxes were near-zero across both ecosystems.

We found a weak negative relationship between CH$_4$ emissions and water table depth in the wetland, with emissions decreasing as the water table approached and flooded the soil surface and a positive relationship between CH$_4$ emissions and the presence of *Sphagnum* mosses. Temperature was also an important driver of CH$_4$ with emissions increasing to a peak at approximately 12°C. Little could be determined about the drivers of N$_2$O emissions given the small magnitude of the fluxes.

A multiple regression modelling approach was used to describe CH$_4$ emissions based on spectral data from PLEIADES PA1 satellite imagery across a 2 x 2 km landscape. When applied across the whole image domain we calculated a CH$_4$ source of 2.05 ± 0.61 mg C m$^{-2}$ hr$^{-1}$. This was significantly higher than landscape estimates based on either a simple mean or weighted by forest/wetland proportion (0.99 ± 0.16 mg C m$^{-2}$ hr$^{-1}$, 0.93 ± 0.12 mg C m$^{-2}$ hr$^{-1}$, respectively). Hence we conclude that ignoring the detailed spatial variability in CH$_4$ emissions within a landscape leads to a potentially significant underestimation of landscape scale fluxes. Given the small magnitude of measured N$_2$O fluxes a similar level of detailed upscaling was not needed; we conclude N$_2$O fluxes do not currently comprise an important component of the landscape scale GHG budget at this site.

# 1 Introduction

Almost a third of the world's soil carbon is estimated to be stored in boreal and sub-arctic wetlands (Gorham, 1991) yet greenhouse gas (GHG) emissions are still poorly constrained (e.g. Bridgham et al., 2013). Furthermore, the potential feedbacks between high latitude carbon and the global atmospheric radiative balance is not fully understood or accurately accounted for in coupled carbon cycle-climate models (Koven et al., 2011). Boreal nitrogen (N) stocks are significantly understudied compared to C. However boreal forests are known to be significant stocks of organic N, peatlands are estimated to contain approximately 10-15% of the global N pool, and permafrost regions are thought to contain between 40–60 Pg of N (Abbott and Jones, 2015;Loisel et al., 2014;Valentine et al., 2006).

It is now accepted that global surface air temperatures are rising and the rate of increase is greatest in these high latitude areas (Pachauri and Reisinger, 2007). Hence understanding both the current magnitude of GHG emissions and the drivers is essential to monitor and predict climate-driven changes and climate feedback mechanisms.

Whilst it is important to understand the direct implications of increased temperature on net GHG emissions ($CO_2$, $CH_4$ and $N_2O$), it is also critical to consider the indirect impact through secondary drivers such as permafrost thaw, changes in vegetation community structure, substrate availability, soil hydrological regimes and flow path dynamics. These factors, both individually and via interactions, are likely to alter both net GHG emissions and GHG speciation; e.g. a recent meta-analysis showed the temperature sensitivity of $CH_4$ was greater than that of $CO_2$ suggesting increased temperature may lead to changes in the $CH_4$:$CO_2$ emission ratio (Yvon-Durocher et al., 2014). The sensitivity of $CH_4$ fluxes to these environmental controls is not currently well understood, for example previous studies show differing responses to water table dynamics (e.g. Aerts and Ludwig, 1997; Olefeldt et al., 2013; Turetsky et al., 2014; Waddington et al., 1996). This limits the ability of mechanistic models to accurately simulate actual net fluxes. A significant research focus is required to fully explain the drivers of GHG emissions to provide a solid basis for future prediction.

Much of the previous research effort in this field has been focussed on $CO_2$, the most abundant atmospheric GHG, often followed by $CH_4$ and to a much lesser extent $N_2O$. Knowledge of the distribution of $N_2O$ fluxes across high-latitude ecosystems is in fact almost entirely lacking. Examples of mean growing season net ecosystem exchange values across subarctic/boreal regions include an uptake of 1.7 g $CO_2$ $m^{-2}$ $d^{-1}$ (Lafleur, 1999) and 5.47 g $CO_2$ $m^{-2}$ $d^{-1}$ (Fan et al., 1995), both from Canadian forest sites, and an uptake of 3.86 g $CO_2$ $m^{-2}$ $d^{-1}$ (Aurela et al., 2002) from a Finnish mesotrophic flark fen. Whilst $CH_4$ and $N_2O$ emissions are generally lower than net $CO_2$ emissions, the greater radiative forcing they produce, as described by the global warming potential (GWP), justifies their inclusion in GHG studies. The 100 year GWPs with and without climate-carbon feedbacks, respectively, are currently estimated as 28 and 34 for $CH_4$ and 265 and 298 for $N_2O$ (Myhre et al., 2013). Overall, boreal forests appear to be a small sink for $CH_4$ and a small source of $N_2O$ (Moosavi and Crill, 1997; Pihlatie et al., 2007) whilst wetlands typically represent sources of $CH_4$, and a small sink for $N_2O$ (e.g. Bubier et al., 1993; Drewer et al., 2010b; Huttunen et al., 2003). Growing season emissions of $CH_4$ from subarctic and boreal ecosystems are estimated as 112.2 $\pm$ 6.2 and 72.7 $\pm$ 1.3 mg $m^{-2}$ $d^{-1}$, respectively (Turesky et al. 2014), compared to modelled estimates of $N_2O$ emissions from

tundra, forest tundra and boreal forest of 0.02, 0.09 and 0.15 mg m$^{-2}$ d$^{-1}$, respectively (Potter et al. 1996). Most studies focus primarily on growing season fluxes. Whilst the logistics of making winter measurements in these ecosystems certainly plays an important role, the growing season has also been shown to represent the period of greatest emissions and therefore the most suitable time to study drivers. Jackowicz-Korczyński et al. (2010) found that summer season CH$_4$ emissions represented 65%

of the annual flux (with the shoulder seasons representing 25% and the winter season only 10% of annual flux) in a subarctic peatland. Similarly Panikov and Dedysh (2000) found that winter CH$_4$ emissions contributed only 3.5 to 11% of total annual flux in Western Siberian boreal peat bogs and Dise (1992) reported winter CH$_4$ fluxes representing between 4 and 21% of total annual flux in peatlands across Northern Minnesota.

Net CH$_4$ emissions are controlled by the balance of activity between anaerobic methanogenic and oxidizing aerobic

methanotrophic bacteria. Hence the degree of soil saturation, which controls the position of the oxic-anoxic boundary and the associated soil redox potential, has been identified as an important driver of net CH$_4$ emission (Bubier et al., 1995; Kettunen et al., 1999; Nykanen et al., 1998). Other factors such as temperature, substrate availability, soil porosity and pH are also commonly reported drivers of CH$_4$ emissions (Baird et al., 2009; Dinsmore et al., 2009b; Levy et al., 2012; Strack et al., 2004; Yvon-Durocher et al., 2014). Whilst the rate of methanogenesis and methanotrophy are both influenced by temperature,

methanogenesis is generally considered to be more temperature-sensitive resulting in a positive relationship between temperature and net CH$_4$ emission (Dunfield et al., 1993; van Hulzen et al., 1999). CH$_4$ produced within the soil environment is then transported to the atmosphere via diffusion, ebullition or plant-mediated transport.

Vegetation can exert either a direct control on CH$_4$ emission via plant-mediated transport, or indirect control via its contribution to soil structure, moisture, anaerobic microsites and substrate availability. The development of aerenchyma is an adaptation to

waterlogged conditions found in many vascular wetland species. Where such species are present they can act as gas conduits, allowing GHGs produced in the anoxic layer to be transported to the atmosphere with minimal oxidation, subsequently increasing emissions by up to an order of magnitude (Dinsmore et al., 2009a; MacDonald et al., 1998; Minkkinen and Laine, 2006). Vegetation community structure also provides a useful proxy for environmental variables that are themselves difficult to measure, such as long-term water table dynamics (Gray et al., 2013; Levy et al., 2012).

The primary processes controlling N$_2$O emissions from soils, including boreal soils, are nitrification processes, where ammonium is oxidised to nitrate under aerobic conditions and denitrification processes, where oxidised nitrogen species are reduced to N$_2$O or N$_2$ under anaerobic conditions (Firestone and Davidson, 1989). As CH$_4$, also N$_2$O production is a microbial process. The main drivers regulating N$_2$O production are nitrogen, such as ammonium and nitrate, temperature and factors which regulate the ratio of aerobic to anaerobic soil microsites, such as soil moisture (Butterbach-Bahl et al., 2013). In

peatlands transport through aerenchyma, is also for N$_2$O a potential transport mechanism.

A number of different in-situ methods are available for the measurement of GHG emissions. Eddy covariance methods can produce high temporal resolution measurements integrated at the field and ecosystem (Baldocchi et al., 2001; Hargreaves and Fowler, 1998); whilst useful for field scale quantification, the method does not allow separation of individual landscape components. Traditional chamber based studies allow a more targeted experimental design where individual

microtopographical features or vegetation communities can be selected and compared (Dinsmore et al., 2009b; Drewer et al., 2010a). By explaining small-scale spatial variability we can gain a greater understanding of GHG drivers and begin to predict how climate or land-use management changes will alter the GHG balance over the full landscape. Furthermore both $CH_4$ and $N_2O$ can be measured simultaneously within the same chamber allowing greater confidence in comparisons between the flux estimates.

There exists a fundamental mismatch between the scale of measurements required to increase process level understanding of $CH_4$ and $N_2O$ emissions, and the scale required to make useful assertions about the magnitude of emission sources that are relevant to the global GHG budget. Whilst land-surface models provide one way to bridge this mismatch of scale, they are often limited by the availability of specific input variables e.g. water table depth, which cannot be measured at the spatial resolution required to provide an accurate output. As a result, modelled estimates of northern high-latitude wetland $CH_4$ sources are highly variable between studies ranging from approximately 20 - 157 Tg $CH_4$ $yr^{-1}$ (Zhu et al., 2013 and references therein). An alternative method of upscaling is empirically mapping emission factors onto spectral data provided by high resolution satellite imagery. This method utilises the spectral signatures of different vegetation types and vegetation specific differences in GHG emissions to create a landscape scale emission ma

In this study we use static-chambers and satellite imagery to assess the primary spatio-temporal drivers of $CH_4$ and $N_2O$ emissions in sub-arctic/boreal Finland and upscale this to a 4 $km^2$ landscape containing both forest and wetland ecosystems.

## 2 Methods

### 2.1 Site Description

The Arctic Research Centre of Sodankylä (67°22'N 26°39'E, 179 m a.s.l.) is located in central Lapland, Northern Finland, approximately 100 km north of the Arctic Circle. The centre is run by the Finnish Meteorological Institute, is part of the Pallas-Sodankylä GAW station and includes a level 1 ICOS ecosystem station. Whilst referenced as an Arctic site in respect to stratospheric meteorology and geographical location, it is considered to be within the sub-arctic/boreal vegetation zone and is not underlain by permafrost. Mean annual temperature and precipitation on site from 1981-2010 was -0.4°C and 527 mm, respectively. Records of mean annual air temperature on site have shown an increase of 0.02°C $yr^{-1}$ over the period 1961-2000; the rate of increase specifically during March to May was 0.04°C $yr^{-1}$ (Aurela et al., 2004; Tuomenvirta et al., 2001). The mean snow depth (mid-March) is 75 cm with median snowfall start and end dates of 26[th] September and 14[th] May (Finnish Meteorological Institute). Permanent snow cover starts approximately end of October, beginning of November. Scots pine forests and wetlands are the two dominant ecosystems in this region. Both ecosystems were covered by the greenhouse gas flux measurements in order to enable the landscape scale upscaling of the results.

The forest (N67°21.708' E26°38.290', 179 m.a.s.l.) is classified as an Uliginosum-Vaccinium-Empetrum (UVET) type Scots pine (*Pinus sylvestris*) forest on a sandy podzol. The mean vegetation height within the forest is 12 m in the area where our measurements were made with an average stand age of 60-100 years and tree density of 2100 $ha^{-1}$. The forest floor contains a

varying degree of lichen (*Cladonia* spp.) which is heavily dependent on the presence/absence of reindeer. We located static chambers evenly between 3 forest sites (unfenced, 12 year enclosure, 50 year enclosure) to ensure variability in GHG emissions due to lichen cover was included in our results. The nearby Halssiaapa wetland (N67°22.111' E26°39.269', 180 m.a.s.l.) is described as a eutrophic fen dominated by large, treeless flarks with abundant sedge vegetation and intermittent brown moss

and *Sphagnum* cover. Intermediate, low ridges consist of birch fen vegetation interspersed with pubescent birch trees (*Betula pubescens*), with a dominant height of approximately 5-7 m. The most common shrubs are *Betula nana*, *Andromeda polifolia* and *Vaccinium oxycoccos*, herbaceous plants are primarily *Potentilla palustris* and *Menyanthes trifoliata*, and grasses are predominantly *Carex* species (several different species observed) or *Scheuchzeria palustris*. Across the duration of the study water table levels varied substantially and no consistently submerged areas which could be easily distinguished as flarks

existed, we have therefore avoided subjective classification of ridges and flarks within our plots and refer only to measurable environmental variables such as water table depth and temperature.

When set within a wider 2 x 2 km landscape unit (to which we will upscale measurements), the proportion of wetland to forest was almost 2:1 with wetlands making up 61% of the area, and forests 32%. The remaining 7% included open water and grass, bare soil and buildings primarily associated with the Sodankylä Arctic Research Centre. Within the larger regional area

described in an associated study by O'Shea et al. (2014) (http://www.eea.europa.eu/data-and-maps/data/corine-land-cover-2006-raster) forests made up a much greater proportion of the landscape with coniferous and mixed forests representing 33% and 16% of the land area, respectively, and wetlands 23%.

## 2.2 Field Methodology

Measurements were carried out during growing season 2012 in two measurement campaigns (summer: 12th July – 2nd August;

autumn: 22nd September – 14th October) with the intention of capturing peak summer $CH_4$ emissions and the subsequent shoulder season.

A total of 60 static chambers were measured, 21 within the forest and 39 within the wetland. Within the forest, 7 chambers were located in each of three subplots representing no enclosure, 12 year enclosure (built in summer 2000) and an approximately 50 year enclosure. Within the wetland, chambers were strategically located to cover the perceived range of

both vegetation communities and water table depths covering both hummock and hollow microtopographic types (chamber numbers per microtopographic type: hummocks = 16, hollows = 11, neither hummock nor hollow = 12). Fluxes were measured on approximately 2 day intervals resulting in a total of 10 measurements for all chambers during the summer campaign, and 7 for the forest and 8 for the wetland chambers during the autumn campaign.

Static chambers were constructed from 40 cm diameter opaque polypropylene pipe following the guidelines discussed in

Clough et al. (2015). Wetland chambers were located so that sampling could be carried out from an existing boardwalk, this served the dual purpose of avoiding disturbance during chamber enclosure and minimised the environmental impact of footfall on the site. The ground surface within the forest plots was considered to be solid and therefore no such precautions were required. Shallow bases (10 cm depth) were inserted into the ground the day before the first sampling; bases were left in-situ

for the remainder of the study period. Fluxes calculated from the first sampling day were not significantly different from subsequent sampling occasions. The short settling period after base installation is therefore considered to have had no significant effect on subsequent fluxes, which therefore were included in the data analysis. Chamber lids, consisting of a 25 cm section of polypropylene pipe with a closed metal top, pressure compensation plug and draft excluder tape for sealing,

were attached and sealed to the in-situ bases during the 45 min flux measurement period. Chamber air (100 ml) was sampled 4 times throughout the approximately 45 minute sampling period and flushed through 20 mL glass vials sealed with butyl rubber plugs using a double needle system; vials were kept at atmospheric pressure reducing problems associated with pressure changes during transportation. Vials were returned to the laboratory at the Centre for Ecology and Hydrology, Edinburgh, for analysis within approximately one month. Samples were analysed on an HP5890 Series II gas chromatograph (Hewlett Packard

(Agilent Technologies) UK Ltd, Stockport, UK) with electron capture detector (ECD) and flame ionisation detector (FID) for $N_2O$ (detection limit<7 μg l−1) and $CH_4$ analysis (detection limit<70 μg l−1), respectively. Soil temperature was recorded at a depth of 10 cm from four replicate points immediately outside the chamber bases on each sampling occasion using the Omega HH370 temperature probe (Omega Engineering UK Ltd., Manchester, UK). Within the forest plots, 4 replicate volumetric soil moisture content (VMC) measurements were made, adjacent to each chamber base, using a Theta probe HH 2 moisture meter

(Delta T-Devices, Cambridge, UK). Within the wetland, a total of 21 dip wells constructed from 5 cm internal diameter pipe, were installed either adjacent to, or where chambers were located close together, between chamber bases. All wetland chambers had at least 1 dip well located within a 50 cm radius, where more than one dip well was located equidistant from the chamber, the mean water table depth from the adjacent dip wells was calculated. Soil respiration (note whilst we refer to this as soil respiration throughout, it also includes respiration from the ground surface vegetation defined as anything with a height of less

than 2 cm above ground surface) was measured using a PP-Systems SCR-1 respiration chamber (10 cm diameter) attached to an EGM-4 infrared gas analyser (IRGA, PP Systems; Hitchin, Hertfordshire, England) on each sampling occasion. Soil respiration was measured adjacent to each forest chamber and adjacent to 14 chambers within the wetland, chosen to cover the perceived range of spatial variability. Vegetation within each chamber was recorded upon visual inspection.

A pair of cation and anion Plant Root Simulator (PRS)[TM] probes were deployed adjacent to each of the 60 chamber bases

during both sampling campaigns. The PRS probes utilise ion-exchange resin membranes to provide an index of relative plant nutrient availability (Hangs et al., 2002), measured ions included total N, $NO_3$-N, $NH_4$-N, Ca, Mg, K, P, Fe, Mn, Zn, B, S, Pb, Al, and Cd. During the summer campaign probes were deployed on the 11th and 12th July, and recovered on the 1st August. During the autumn campaign forest probes were deployed on the 22nd and 23rd September and recovered between the 13th and 15th October. As part of the standard analytical processing, concentrations from each probe are corrected for length of

deployment. After recovery, probes were processed and cleaned with deionised water following the standard procedure supplied by the manufacturers and returned to Western Ag Innovations Inc., Canada for analysis.

## 2.3 Data Analysis

Fluxes and confidence intervals from static chambers were calculated using GCFlux, version 2, which calculates fluxes based on 5 methods before choosing the most appropriate fit for individual chamber sets (Levy et al., 2011). Reported $CH_4$ fluxes correlate to the best-fit model for individual chambers (either linear or asymptotic). Due to the larger uncertainty in calculated

$N_2O$ concentrations which are often close to the GC detection limits, reported $N_2O$ fluxes were calculated from the linear model approach only. Instantaneous fluxes are presented in units of nmol $m^{-2}\,s^{-1}$. Confidence intervals include errors introduced by a combination of natural variability in the flux over the measurement period, methodological and analytical limitations and uncertainty in model fitting. When these range from negative to positive, no sign can be accurately attributed to the flux and therefore it is treated as indistinguishable from zero.

The data distribution of fluxes, from all chambers, and over the full study period, had a strong positive skew (Figure 1). To summarise the data and account for the skewed distributions, geometric means were calculated across time points for all chambers. Where periods of uptake and emission were both present within a time series, geometric means were calculated for each flux direction independently. The presented geometric means are the frequency-weighted sum of emissions and uptake. The resulting spatial dataset had a distribution much closer to normal and is therefore summarised throughout using arithmetic

means. Upscaled emission estimates are presented in units of either g C $m^{-2}\,hr^{-1}$ or g N $m^{-2}\,hr^{-1}$ for $CH_4$ and $N_2O$, respectively. In all further analysis, log transformations were applied where data-sets displayed non-normal distributions; given the time between measurements, autocorrelation within datasets was never significant. To summarise the complex vegetation and soil data, principal component analyses (PCA) were performed using the princomp function within the R stats package (R version 3.1.1), this uses a spectral decomposition approach which examines the covariances and correlations between variables.

Correlation analyses were carried out with principal components one, two and three (PC1, PC2, PC3) against spatial $CH_4$ fluxes and the most appropriate component taken forward into subsequent explanatory models. No attempt to correlate vegetation or soil components was made with $N_2O$ fluxes given the large proportion of near-zero fluxes.

Spatial variability between chambers on all sampling occasions was large. To allow temporal variability to be considered it was necessary to group chambers. Rather than subjectively assign chambers to groups based on observed landscape features

we carried out a cluster analysis (R, version 3.1.1) based on emission rates. This method produced independent groups which could also be used in further analyses to consider the environmental controls of emissions. The total number of clusters was chosen to be 5, after multiple cluster analysis runs this was considered the most appropriate number taking into consideration the complexity for further analyses and clear distinctions between groups. ANOVA and Tukey's pairwise comparisons were used to explore the differences in environmental variables between clusters, tested variables included means of soil

temperature, water table depth and soil respiration alongside vegetation principal component and soil principal component.

Optical remote sensing imagery was acquired by the Pleiades satellite on 28th August 2012. This provided data in the blue, green, red, and near-infrared (NIR) part of the spectrum for the 2 x 2 km region around the chamber sites, with 2 m resolution on the ground.  From these data both the simple ratio (SR = NIR / Red) and normalised difference vegetation index (NDVI =

[NIR - Red] / [NIR + Red]) were calculated. The optical data for each chamber location were extracted and related to the geometric mean of the $CH_4$ flux at that location. Multiple regression modelling was then carried out using R (version 3.1.1) to describe the $CH_4$ fluxes of individual chambers initially utilising all four wavebands and the two calculated ratios. The best fit model was used to upscale $CH_4$ fluxes to the full image domain (4 km$^2$). Due again to large uncertainties in the flux estimates, large proportion of fluxes indistinguishable from zero, and subsequent inability to accurately model the data, upscaling of $N_2O$ emissions was not carried out using satellite imagery.

## 3 Results

Confidence intervals calculated from each chamber measurement, which include errors introduced by a combination of natural variability in the flux over the measurement period, methodological and analytical limitations and uncertainty in model fitting, show a high proportion of calculated fluxes which are indistinguishable from zero. Given the high relative variability in individual chambers and low fluxes in $N_2O$, only 8 and 9 % of fluxes were significant in the wetland and forest, respectively. The proportion of chambers displaying significant $N_2O$ fluxes could not be linked to any measured environmental factors and were distributed randomly across the dataset. For $CH_4$, whilst only 56% of fluxes were significantly different from zero in the forest, the wetland was much clearer with zero excluded from the confidence range in 94% of cases.

When separated by site (forest, wetland) and by campaign period (summer, autumn) the highest instantaneous $CH_4$ fluxes, greatest skew and largest range were all observed in the wetland chambers during the summer period (Figure 1). These equated to a mean flux of $3.35 \pm 0.44$ mg C m$^{-2}$ hr$^{-1}$, compared to only $0.62 \pm 0.09$ mg C m$^{-2}$ hr$^{-1}$ in the wetland during the autumn period. The mean $CH_4$ flux across the whole measurement period represented an emission of $1.56 \pm 0.20$ mg C m$^{-2}$ hr$^{-1}$ from the wetland chambers, compared to a mean uptake of $0.04 \pm <0.1$ mg C m$^{-2}$ hr$^{-1}$ from the forest chambers (Table 1).

$N_2O$ fluxes had a mean emission across the full sampling period of $1.06 \pm 0.44$ µg N m$^{-2}$ hr$^{-1}$ and $0.73 \pm 0.40$ µg N m$^{-2}$ hr$^{-1}$ from forest and wetland chambers, respectively (Table 1).

## 3.1 Spatial Variability

Surface cover data (vegetation and presence of standing water) was summarised using a PCA analysis; combined the top three principal components explained 51% of the total variation between chamber vegetation communities, with principal components 1, 2 and 3 (PC1, PC2, PC3) explaining 24%, 15% and 11%, respectively. PC1, PC2 and PC3 were subsequently tested for correlations with $CH_4$ fluxes. Spatial variability in $CH_4$ emissions among wetland chambers was best captured using PC2 ($r = 0.40$, $P < 0.01$). PC2 also correlated strongest with $CH_4$ emissions when all chambers (both wetland and forest) were included ($r = 0.31$, $P < 0.01$), however PC1 showed the best correlation with forest chambers alone ($r = 0.25$, $P < 0.01$). PC2 was therefore used throughout future analysis to describe the spatial variability in $CH_4$ emissions.

PC2 (which best described $CH_4$ fluxes) showed a strong dependence on the proportion of green *Sphagnum* species within the chamber with positive PC2 values indicating a high prevalence (Figure 2). Due to the strongly non-normal distribution of the

data, *Sphagnum* sp. alone could not be correlated with emissions, thus the principal component method provides an indirect measure of the relationship. Low PC2 scores indicate a higher abundance of non-*Sphagnum* moss species and high proportion of open water within the chambers. Of the measured environmental variables relating to spatial variability (soil temperature, soil moisture, water table depth and soil respiration), PC2 only correlated significantly with water table depth ($r = 0.17$, $P <$

0.01) with PC2 scores increasing with water table depth.

A similar PCA analysis was carried out to summarise the available soil nutrient availability data from the PRS probes. The first three principal components combined explained 56% of total variation with PC1, PC2 and PC3 individually accounting for 31%, 15% and 10% of variability, respectively. PC1 gave the best correlation with $CH_4$ emissions when all data was combined and for forest chambers alone. PC2 gave a better correlation with wetland chambers alone (PC2: $r = 0.40$, $P < 0.01$).

PC2 was therefore utilised throughout the remainder of the analysis due to the greater magnitude of wetland versus forest $CH_4$ emissions, and their subsequent importance to landscape scale emissions.

PC2 was influenced strongly by total N and $NH_4^+$ concentrations with high concentrations resulting in a low PC2 score (Figure 3). The only environmental variable significantly correlated with PC2 was water table depth ($r = 0.19$, $P < 0.01$) with high PC2 scores indicating a deep water table. However, when wetland chambers were considered alone soil respiration also showed a

significant positive correlation with PC2 ($r = 0.31$, $P < 0.01$).

Spatial variability in GHG emissions were tested against the measured environmental variables as well as the most appropriate PCA score for both vegetation and soil, as described above. $CH_4$ flux was not statistically correlated to water table depth in the wetland chambers (Figure 4). However a relatively strong positive correlation was seen between $CH_4$ flux and the PCA score from the vegetation analysis; a high score from the vegetation principal component represented a deep mean water table depth.

Positive correlations were also found between $CH_4$ flux, mean soil temperature and the principal component from the soil analysis when the wetland chambers were considered alone. Within the forest chambers, only the soil principal component was statistically correlated to $CH_4$ flux.

To further summarise the $CH_4$ data and provide a method for both upscaling and consideration of temporal variability, chambers were grouped independently based on net emissions. Data distributions within each cluster group are shown in Figure

5. The cluster identified with the lowest emissions contained all the forest chambers and an additional two low emitting wetland chambers; for explanatory purposes this cluster is subsequently referred to as the 'forest' cluster. The remaining clusters, with sequentially increasing emissions, are labelled wetland_a, wetland_b, wetland_c and wetland_d, respectively.

ANOVA showed significant between cluster variability in all tested environmental variables (soil temperature, water table depth, soil respiration, vegetation principal component and soil principal component) with the exception of water table depth

(Figure 6). The patterns in soil temperature, PCA_veg and PCA_soil are in line with the previously discussed correlation analysis. When the components of PCA_veg are considered independently the results highlight the importance of *Sphagnum* cover and open water in controlling the $CH_4$ emissions within the wetland clusters, however this relationship is complicated by the high variability shown by large standard deviations from the mean cluster values (Table 2). Wetland clusters 'a' and

'b', which represent the two lowest emitting wetland groups, had the lowest proportions of *Sphagnum* moss species and the greatest proportion of chambers containing open water.

Between-group differences in soil nutrient concentrations were also considered using ANOVA; only nutrients which displayed significant between-group differences are displayed in Figure 7. The strongest between-group difference was evident in the soil Fe concentrations, with high Fe linked to high $CH_4$ emitting chambers ($F = 62.0$, $P < 0.01$); positive correlations with mean group $CH_4$ emissions were also seen for B ($F = 49.2$, $P < 0.01$), Zn ($F = 39.0$, $P < 0.01$) and Mg ($F = 49.2$, $P < 0.01$). Negative correlations were seen between mean group $CH_4$ emission and K ($F = 10.6$, $P < 0.01$), $NO_3$-N ($F = 6.38$, $P < 0.01$), and $NH_4$-N ($F = 6.36$, $P < 0.01$). Within the wetland, total-N was lowest in groups with the highest $CH_4$ emission; however the pattern is less clear when forest chambers are included as these displayed a wide range of total-N but a low $CH_4$. Only the forest had distinct soil Ca concentrations.

## 3.2 Temporal Variability

Temporal variability, summarised by cluster, is displayed in Figure 8 for both the summer and autumn campaign periods. $CH_4$ emissions remain relatively constant throughout both campaign periods despite a significant drop in emissions between them. Despite the low temporal variability, emissions appear to peak around mid-July in the higher emitting chamber clusters (e.g. wetland_c and wetland_d).

$CH_4$ emissions did not follow linear relationships with the measured environmental variables (soil temperature, air temperature, water table depth and soil respiration) (Figure 9). $CH_4$ emissions peaked at a soil temperature of approximately 12°C and an air temperature of approximately 15°C, after which they began to fall. The time series suggests a general decrease in $CH_4$ emissions with rising water table, however the relationship appears to be chamber specific and non-linear suggesting a greater complexity than is usually accounted for. In the high emitting chambers, there is a peak in $CH_4$ emissions as the water level reaches the surface, the emissions drop until water tables of approximately 5 cm depth and then rise again as the water level deepens further. Chamber clusters associated with lower total $CH_4$ emissions did not show this peak associated with surface water tables but instead followed a smoother, but still non-linear, increase in emissions with increasing water table depth. No relationship was observed between soil temperature and water table depth ruling out a potential interaction as the cause of the peaks associated with particular water table depths or soil temperatures.

## 3.3 Spectral analysis and upscaling

A multiple regression model including blue, green, red, NIR, SR and NDVI explained 45% of the variance in the spatial $CH_4$ flux. Transformations of the data and more complex models were explored, but did not substantially improve the model fit. A simpler model containing only SR, NDVI and the blue and NIR wavebands performed equally as well as the full model also explaining 45% of the spatial variation (Table 3), this simpler model was therefore used in subsequent analysis. To predict mean $CH_4$ flux over our sampling period at landscape scale, we applied the regression model to the optical data over the whole 2 x 2 km domain. This predicted high $CH_4$ fluxes in the wetland areas in the north-east and at forest edges (Figure 10). Using

the optical data to scale up the chamber measurements, the mean $CH_4$ flux over the whole domain between $12^{th}$ July and $14^{th}$ October is estimated to be $47.4 \pm 14.1$ nmol $CH_4$ m$^{-2}$ s$^{-1}$ or $2.05 \pm 0.61$ mg C m$^{-2}$ hr$^{-1}$. By comparison, if the flux over the whole spatial domain were estimated simply as the arithmetic mean of the individual chamber measurements (geometric mean to summarise temporal variability) the value would be significantly lower ($23.0 \pm 3.78$ nmol $CH_4$ m$^{-2}$ s$^{-1}$). If we account for the

differences between wetland and forest alone using an appropriate area weighting factor (61% wetland; 32% forest), ignoring variability within these landscape units, estimated emissions are $21.6 \pm 2.85$ nmol $CH_4$ m$^{-2}$ s$^{-1}$, also substantially lower than our modelled approach.

## 4 Discussion

Fluxes of $CH_4$ from the forest and wetland areas within the landscape were significantly different at $-0.06 \pm <0.01$ and $3.35 \pm$

$0.44$ mg C m$^{-2}$ hr$^{-1}$, respectively. Whilst the error displayed here suggests confidence in the forest as a net sink for $CH_4$, when individual chamber measurements are considered, only 56.3% of the measured fluxes had an error bar that did not cross the zero line. Hence we can only be confident that the sign of the flux is correct in just over half of our forest data. On removal of all fluxes with an uncertain sign, the mean remains negative in the forest chambers. This gives confidence that whilst the calculated flux is very small, it is a small sink rather than a source. In the wetland however, 94.4% of the measured fluxes

differed significantly from zero, so we can be confident that the wetland represented a strong source of $CH_4$.
A similar analysis was carried out on the $N_2O$ flux data and here due to very high uncertainties in the sign of individual flux measurements (only 8.68% and 7.79% of measurements in the forest and wetland, respectively, did not have error bars crossing the zero line) we cannot differentiate either the forest or wetland as being a net sink or source over the campaign period. We can simply state that $N_2O$ fluxes in both landscape units were near-zero. $N_2O$ fluxes were therefore not an important component

of this study area. Whilst minimal, the near-zero result is still an important finding given the lack of $N_2O$ emissions reported in the current literature. Assuming these near-zero fluxes are similar across the region we have an important baseline from which to monitor change related to future climate or land-use practices. However, due to consistently near-zero fluxes little could be concluded about the drivers of $N_2O$ emissions within our landscape area.

### 4.1 Drivers of $CH_4$ emissions

The relationship between $CH_4$ emissions and water table position was not straightforward. Considering the mean $CH_4$ flux for each chamber and testing this against the mean water level position of that chamber showed no significant relationship (Figure 4), suggesting water table was not an important factor in controlling spatial variability in emissions across the site. Furthermore, when chambers were clustered based on their $CH_4$ emissions, there was high within-group variability in water table and subsequently no significant differences in water table between groups (Figure 6). Whilst much of the previous literature

suggests water level as the primary driver of $CH_4$ (Aerts and Ludwig, 1997; Hargreaves and Fowler, 1998; Waddington et al., 1996) due to its role in controlling the oxic/anoxic boundary, there is a growing body of evidence which suggests this is true

only in drier ecosystems (Hartley et al., 2015; Olefeldt et al., 2013; Turetsky et al., 2014). The water levels used in this analysis only represented the water level during the campaign periods, with no consideration of longer term means. Due to the presence of alternative electron acceptors and the delay in returning to favourable redox conditions, fluctuations in the water level can result in a reduced population and a subsequent reduction in $CH_4$ production, even after water levels and anoxic conditions

recover (Freeman et al., 1994; Kettunen et al., 1999). Hence whilst soil conditions may appear suitable for $CH_4$ production at the time of measurement, an unfavourable water table in the days to weeks prior to the measurement can limit methanogenesis and mask the expected relationship.

$CH_4$ in the wetland correlated positively and significantly with a component from the vegetation PCA analysis. The vegetation component that best described $CH_4$ emissions (PC2) related primarily to *Sphagnum* cover within the chambers and also linked

low scores to a high proportion of open water. *Sphagnum* is an indicator of long term near-surface water table position, hence whilst the directly measured water table did not correlate significantly with $CH_4$ emissions, the vegetation analysis suggests that longer term water level conditions do correlate with spatial variability in $CH_4$. Several other studies have also highlighted the importance of vegetation as an indirect indicator of $CH_4$ flux as it integrates across multiple ecological variables (e.g. Bubier et al., 1995; Davidson et al., 2016; Gray et al., 2013; Oquist and Svensson, 2002). It is also this link to vegetation that

makes upscaling such as that described below possible as the spectral data is primarily picking up spatial variability in above-ground plant community cover. Vegetation can also play an important direct role in GHG emissions via plant-mediated transport and the supply of labile substrate, thought to be particularly important for methanogenesis (e.g. McEwing et al., 2015; Ström et al., 2005). *Sphagnum* mosses can be additionally important in controlling $CH_4$ emissions through their association with methanotrophic bacteria, an association that has been shown to exist across the globe and across a range of

microtopographic features (Kip et al., 2010). Here we find no correlation between $CH_4$ emissions and soil respiration and a positive influence of *Sphagnum* cover. This suggests that the role of vegetation as an indirect indicator of other environmental factors is more important to $CH_4$ emissions in this landscape than methanotrophic associations or substrate availability.

The water table relationship is further complicated by the presence of standing water which related to low emitting chambers. This may be a consequence of reduced diffusion from the soil to the atmosphere rather than a result of reduced production. If

standing water remains for long periods of time, the sustained anoxic conditions can alter the vegetation and soil chemistry. For example reduced nitrification, an oxic process, can lead to a build-up of $NH_4^+$ in water logged conditions. Soil PCA component 2 which correlated positively with $CH_4$ emissions showed a strong link to the concentration of $NH_4^+$; high concentrations were linked to low PCA scores and low $CH_4$ emissions. $NH_4^+$ in this case may be acting as an indicator of the chambers which were inundated with surface water for sustained time periods.

Our chambers were not specifically designed to measure emissions from water surfaces and as a result cut out all wind driven turbulence which is likely to be an important driver of the evasion flux (MacIntyre et al., 1995). It is therefore difficult to identify whether standing water produced a decrease in $CH_4$ production, a real decrease in flux due to low diffusivity through the water column, or if our results were a consequence of our methodology artificially reducing gas transfer across the water-air boundary. A previous study showed an increase in $CH_4$ emissions along a water table gradient from 35 cm depth to 5 cm

above the soil surface. Above 5 cm the relationship with increasing water level was negative (Pelletier et al., 2007). Whilst our results are not as clear as those presented by Pelletier *et al.* (2007) a similar mechanism of reduced $CH_4$ diffusion through standing water may be responsible in both cases.

Figure 7 shows a clear positive relationship between Fe, Zn and $CH_4$ emissions, with high emitting clusters also displaying the highest concentrations. These cations reflect the redox potential of the soil with increasing concentrations indicating a lowering of the redox potential. The $CH_4$ water table relationship is indirect, with water table used as a proxy for soil oxygen content and redox potential. Here we find cation concentrations have a greater explanatory power than water table hence they may represent a more appropriate indicator of soil redox status and methanogenic potential.

When we consider the temporal patterns in $CH_4$ emissions across the 2 campaign periods we see a similar response as in the spatial analysis, with emissions falling as the water level rises between approximately 15 and 5 cm depth. No relationship was found between water table and soil temperature ruling out an interaction as the primary cause of the water table relationship. Tupek *et al.* (2014) measured increasing $CH_4$ emissions in response to a rising water table until a peak at approximately 20 cm depth in a central Finnish mire, after which the relationship changed with emissions decreasing as the water table approached the surface. Water table depths measured in this study covered a smaller range and therefore we can assume similar dynamics may be apparent if the water level was to drop below 20 cm. Similarly a recent synthesis (Turetsky et al., 2014) involving 71 wetlands found the optimum water table depth for $CH_4$ emissions to be $23.6 \pm 2.4$ cm for bog ecosystems, again suggesting the negative water table relationship observed here is due to water table depth being consistently above the optimum. Potential explanations for the inhibition of $CH_4$ emissions at high water levels given by Turetsky et al. (2014) include limited diffusion of $CH_4$ through standing water as discussed above, reduced $CH_4$ production due to lower plant biomass and associated labile C inputs, or unfavourable redox conditions resulting from inputs of oxygen rich water potentially containing alternative electron acceptors. Whilst we saw no clear correlations between the percentage of bare soil and that of open water in our chambers, a reduction in plant activity may have occurred during submersion so reduction in C inputs for methanogenesis cannot be ruled out to explain the temporal changes in $CH_4$ emissions across the growing season. Neither do we have the data to rule out a change in redox potential due to water flow. A more detailed analysis under controlled conditions would be required to accurately explain the mechanism for high water $CH_4$ limitation at this site.

As the water table rose between 5 cm depth and the soil surface, emissions appear to increase again peaking at approximately the soil surface and then decreasing with increasing water depth above the soil surface. This could be due to physical forcing of $CH_4$ out of the soil pore space as it reaches the soil surface. Importantly, what our results clearly show is that there are a number of driver mechanisms interacting to produce the observed $CH_4$-water table relationship.

A significant positive spatial relationship was seen between soil temperature and $CH_4$ (Figure 4 and Figure 6). The relationship between $CH_4$ emission and temperature is a well-established one often observed in the literature (Segers, 1998) as a result of the greater sensitivity of methanogenesis than methanotrophy; however most studies focus on the implications of temporal variation rather than the spatial pattern. The spatial variability in soil temperature is likely to be linked to a combination of soil

water content and the surface reflectance of the vegetation cover. Changing soil temperature therefore represents an important by-product of other environmental changes that needs to be accounted for in predictive mechanistic models.

The temporal relationship between $CH_4$ emissions and temperature showed a Gaussian response curve typical of microbial control. Peak $CH_4$ emission occurred at a soil temperature of ~12°C. A similar pattern was observed in a central Finland mire by Tupek *et al.* (2014) who recorded a peak in emissions corresponding to 14°C.

## 4.2 Upscaling

The wetland $CH_4$ fluxes calculated here (3.35 mg C $m^{-2}$ $hr^{-1}$ during the summer season and 1.56 mg C $m^{-2}$ $hr^{-1}$ when the autumn period is included) are similar in magnitude to those described in a multisite analysis by Turetsky et al. *(2014)* for subarctic (3.51 ± 0.19 mg C $m^{-2}$ $hr^{-1}$) and boreal wetlands (2.27 ± 0.04 mg C $m^{-2}$ $hr^{-1}$). However given the large differences between fluxes calculated within the forest and wetland, and the heterogeneous mix of these two primary ecosystem types across the subarctic/boreal system, landscape scale emissions are of greater importance in understanding global $CH_4$ source estimates than wetland emissions alone. By extending our sampling site to a 2 x 2 km landscape we can calculate emissions which are more relevant to the region as a whole. Based on a weighted average of fluxes from the forest and wetland within the landscape, and assuming $CH_4$ emissions from the other landscape units are zero, we can calculate average landscape scale emissions of 0.93 ± 0.12 mg $CH_4$-C $m^{-2}$ $hr^{-1}$.

Whilst calculations at this level of detail have previously been shown to give good agreement with more top down methodologies (O'Shea et al., 2014), significant information is lost regarding spatial variability which we have already shown to be large, especially within the wetland. Utilising spectral data across the 2 x 2 km landscape and a multiple regression model, we calculated average $CH_4$ flux over the growing season as 47.4 ± 14.1 nmol $CH_4$ $m^{-2}$ $s^{-1}$ or 2.05 ± 0.61 mg C $m^{-2}$ $hr^{-1}$. This is significantly higher than the 100 $km^2$ landscape scale $CH_4$ flux of 1.1 to 1.4 g $CH_4$ $m^{-2}$ during the May to October growing season (0.19 to 0.23 mg C $m^{-2}$ $hr^{-1}$) calculated by Hartley et al. (2015). The Hartley et al. (2015) study was based on field measurements collected approximately 240 km north of our study site, up-scaled using aerial imagery and satellite data. Even when utilising data presented from only July-September, Hartley et al. (2015) still recorded much lower landscape scale fluxes (approximately 0.24 mg C $m^{-2}$ $hr^{-1}$) than this study due to the different landscape units and proportions of vegetation communities. Whereas we carried out our upscaling over an area characterised by 61% wetlands and 32% forest, the landscape unit measured by Hartley et al. (2015) contained only ~22% wetland (classified as both mire and mire edge) and 60% forest. Heikkinen et al. (2004) also upscaled chamber based $CH_4$ emissions to the landscape, in this instance a 114 $km^2$ catchment in the eastern European Russian tundra, concluding a mean summer $CH_4$ emission rate of 0.43 mg C $m^{-2}$ $hr^{-1}$. $CH_4$ emissions from areas classified as peaty tundra (including intermediate flarks, *Carex + Sphagnum* and hummocks), which ranged from 0.15 to 4.25 mg C $m^{-2}$ $hr^{-1}$, were similar to those presented here. However, again it is the proportion of wetland within the landscape (16.1 %) and to a lesser extent the distribution of emissions within the wetland that appears to be most important in defining the landscape scale flux.

Open water has not been included in this study as it was not an important feature in the 2 x 2 km study area. However given the large proportion of lakes and ponds across sub-arctic and boreal ecosystems, and the potential increase in surface water as the changing climate alters subsurface hydrology, this is something that will become more important both in the future and as we scale to larger or regional landscapes. Methane emissions from Arctic lakes are estimated to total 11.86 Tg yr$^{-1}$, varying

spatially over high latitudes from 3.46 mg C m$^{-2}$ hr$^{-1}$ in Alaska to 0.40 mg C m$^{-2}$ hr$^{-1}$ in northern Europe (Tan and Zhuang, 2015); this puts lake fluxes in the same order of $CH_4$ emissions as northern high-latitude wetlands and comparable to the values measured in this study.

There is still considerable uncertainty in extrapolating to our 2 x 2 km landscape despite optical remote sensing data having complete coverage and a reasonably well-defined relationship with $CH_4$ flux. Greatest emissions and subsequently the greatest

uncertainty are observed in an area to the north east of our landscape which represents an area of yellow/green *Sphagnum*. Further flux measurements are required to reduce the uncertainty in this area. Therefore, in addition to providing upscaled emission estimates, this spectral approach could also potentially be applied to define specific areas for future research focus, maximising the potential explanatory power of future campaigns. .

We were unable to carry out a similar upscaling exercise for $N_2O$, however given the near-zero fluxes across the majority of

study chambers, a detailed spatial method is not required to say with a large degree of certainty that $N_2O$ emissions are not currently a major component of the growing season GHG balance of our landscape. Whilst potentially subject to changes in temperature and hydrology as a result of climate, our site was not underlain by permafrost and therefore is not going to be affected by thaw-related processes. Recent studies have shown potentially large increases in $N_2O$ emissions related to permafrost thaw (Elberling et al., 2010; Abbott and Jones, 2015) so whilst negligible here, $N_2O$ emissions across the wider

northern boreal and sub-arctic zone may become increasingly important to the total GHG balance of the landscape and should therefore continue to be monitored in future research.

### 4.3 Conclusions

Our results showed a significant proportion of measured $N_2O$ fluxes, across both wetland and forest, and $CH_4$ fluxes within the forest, were not distinguishable from zero. Considering only those fluxes that did differ significantly from zero we can be

confident that the wetland represented a strong source of $CH_4$, especially during the summer peak growing season ($3.35 \pm 0.44$ mg C m$^{-2}$ hr$^{-1}$), and the forest a small $CH_4$ sink (summer: $-0.06 \pm <0.01$ mg C m$^{-2}$ hr$^{-1}$). We conclude that $N_2O$ fluxes were near-zero across the landscape in both forest and wetland. Despite the small magnitude of $N_2O$ fluxes this is still an important result given the current lack of data available for $N_2O$ across northern boreal, sub-arctic regions, and the potential for future increases in relation to climate and land-use.

We did not observe a direct water table control on spatial variability in $CH_4$ emissions but instead found a relationship with vegetation communities, in particular the presence of *Sphagnum* mosses, and with soil chemistry which we attribute to redox potential. Both these parameters suggest that water table level and water table variability over a longer time scale prior to flux measurements is required to accurately predict $CH_4$ emissions. When temporal variability across the campaigns was considered

we found a decrease in $CH_4$ emissions as water table approached the soil surface and the soil became fully saturated. We attribute this apparent reversal of the literature described relationship between $CH_4$ and water table to the water table depth being consistently above the optimum. As water levels continue to rise beyond this point diffusion becomes restricted and the flux diminished. We also found a temporal relationship between $CH_4$ emissions and soil temperature with peak emissions at approximately 12°C.

To upscale the chamber measurements of $CH_4$ to a 2 x 2 km landscape area we utilised PLEIADES PA1 satellite imagery and could account for 45% of spatial variability in $CH_4$ flux using SR, NDVI, Blue and NIR spectral data. Applying this model to the full area gave us an estimated $CH_4$ emission of $2.05 \pm 0.61$ mg C m$^{-2}$ hr$^{-1}$. This was higher than landscape estimates based on either a simple mean or weighted by forest/wetland proportion alone ($0.99 \pm 0.16$ mg C m$^{-2}$ hr$^{-1}$, $0.93 \pm 0.12$ mg C m$^{-2}$ hr$^{-1}$, respectively). Hence whilst there are clearly uncertainties associated with the modelled approach, excluding spatial variability as with the latter two methods is likely to lead to underestimations in total emissions. This approach therefore has considerable potential for increasing the accuracy of future landscape scale emission estimates and making better use of the wide variety of chamber measurements currently presented in the literature. When compared to similar upscaling studies our landscape estimate showed significantly higher $CH_4$ emissions, even when individual chamber scale fluxes were similar. Whilst spatial variability within the wetland area was important, the primary difference was the proportion of ecosystem units within the measurement landscape e.g. the proportion of wetland vs forest or tundra. It is therefore not applicable to take the results presented here and simply apply the landscape mean to larger area given the proportion of wetland will change substantially with scale, however with the addition of further ground-truthing and a larger spectral image, larger areas could be similarly modelled.

**Data availability**

Data will be made available through the Environmental Information Data Centre (EIDC), a Natural Environment Research Council data center hosted by the Centre for Ecology & Hydrology (CEH), UK.

**Author contribution**

U.M. Skiba, designed the field experiment, J. Drewer, K.J. Dinsmore and Ute Skiba and carried out the field experiments and subsequent laboratory analysis. A. Lohila and M. Aurela hosted the field sites, provided help with the field site selection, the experimental set-up and local knowledge. P.E. Levy and C. George analyzed the spectral data and developed the model for upscaling chamber emissions to landscape. K.J. Dinsmore carried out the remainder of the analysis and prepared the manuscript with contributions from all co-authors.

## Acknowledgements

This work was funded through the MAMM project (Methane and other greenhouse gases in the Arctic: Measurements, process studies and Modelling, http://arp.arctic.ac.uk/projects/) by the UK Natural Environment Research Council (grant NE/I029293/1).

We wish to thank staff from the Finnish Meteorology Institute at Sodankylä and Helsinki, in particular Annalea Lohila, Tuula Aalto and Tuomas Laurila for their kind hospitality, invitation and collaboration and the EU project InGOS for supporting the two field campaigns in Sodankylä through the TNA2 travel budget (*www.**ingos**-infrastructure.eu/access/**tna2**-access-to-stations).*

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

**Tables**

Table 1. Mean ± SE $CH_4$ and $N_2O$ fluxes split by both campaign period (summer, autumn) and site (forest, wetland).

|  | Summer | Autumn | Full Period |
|---|---|---|---|
| *$CH_4$ (mg C m$^{-2}$ hr$^{-1}$)* |  |  |  |
| Forest | -0.06 ± <0.01 | -0.03 ± <0.01 | -0.04 ± <0.01 |
| Wetland | 3.35 ± 0.44 | 0.62 ± 0.09 | 1.56 ± 0.20 |
|  |  |  |  |
| *$N_2O$ (µg N m$^{-2}$ hr$^{-1}$)* |  |  |  |
| Forest | 0.75 ± 0.33 | 1.29 ± 1.39 | 1.06 ± 0.44 |
| Wetland | 1.63 ± 0.64 | -1.60 ± 1.18 | 0.73 ± 0.40 |

Table 2. Mean ± stdev ground cover data for wetland clusters. Only variables which showed significant between cluster variability are included. Test statistic refers to the F-value with * and ** indicating P-vales of <0.05 and <0.01, respectively.

|  | *Sphagnum sp.* | Open water |
|---|---|---|
| Wetland_a | 39.3 ± 46.0 | 59.5 ± 62.0 |
| Wetland_b | 68.6 ± 47.4 | 11.4 ± 18.6 |
| Wetland_c | 95.7 ± 11.3 | 5.71 ± 9.32 |
| Wetland_d | 50.0 ± 70.7 | 20.0 ± 28.2 |
|  |  |  |
| *ANOVA test statistic* | 4.62** | 3.59* |

Table 3. Model summary utilising spectral data to estimate $CH_4$ emissions

|  | Estimate | t-value | p-value |
|---|---|---|---|
| Intercept | -233 | 0.00002 | <0.01 |
| SR | 354 | 0.00002 | <0.01 |
| NDVI | -283 | 0.03883 | <0.05 |
| Blue | 0.99 | 0.00365 | <0.01 |
| NIR | -0.91 | 0.00022 | <0.01 |
|  |  |  |  |
| *model adjusted $r^2$* | 0.45 |  |  |
| *model p-value* | <0.01 |  |  |

**Figure Captions**

Figure 1. Frequency plot showing distribution of all fluxes across both campaign periods.

Figure 2. Loading values for principal components 1 and 2 of the chamber vegetation analysis.

Figure 3. Loading values for principal components 2 and 3 of the chamber soil concentration analysis.

Figure 4. Relationships between geometric mean $CH_4$ flux (nmol m$^{-2}$ s$^{-1}$) against measured environmental variables across full sampling period. Text refers to the results from statistical correlations were 'ns' refers to a non-significant results, * and ** represent $P < 0.05$ and $P < 0.01$, respectively.

Figure 5. Chambers clustered based on emissions with n indicating the number of chambers within each group.

Figure 6. Boxplots showing range of measured environmental variables within each of the $CH_4$ clusters. Letters represent results from Tukeys family test statistic where clusters with similar letters are not significantly different from one another at 95% confidence level. Clusters Wetland_a to Wetland_d represent groups with sequentially increasing $CH_4$ emissions. Note water table was not measured within the forest plots therefore the water table values given in the 'forest' cluster actually represent only the 2 wetland chambers which have quantitatively been assigned to this cluster and have therefore been excluded.

Figure 7. Boxplots showing range of soil variables within each of the $CH_4$ clusters. Units represent probe supply rate (µg per 10cm$^2$ across burial period). Letters represent results from Tukeys family test statistic where clusters with similar letters are not significantly different from one another at 95% confidence level. Clusters Wetland_a to Wetland_d represent groups with sequentially increasing $CH_4$ emissions.

Figure 8. Temporal variability across the 2 field campaigns in $CH_4$ emissions, separated by clusters, with shaded area representing loess smoothing. Clusters Wetland_a to Wetland_d represent groups with sequentially increasing $CH_4$ emissions.

Figure 9. Drivers of temporal variability in $CH_4$ fluxes, separated by clusters, with shaded area representing loess smoothing. Clusters Wetland_a to Wetland_d represent groups with sequentially increasing $CH_4$ emissions.

Figure 10. Mean (a) and SE (b) of $CH_4$ fluxes extrapolated over a 2 x 2 km area predicted from chamber flux measurements (black circles), and satellite spectral data. Coordinates are in WGS84.

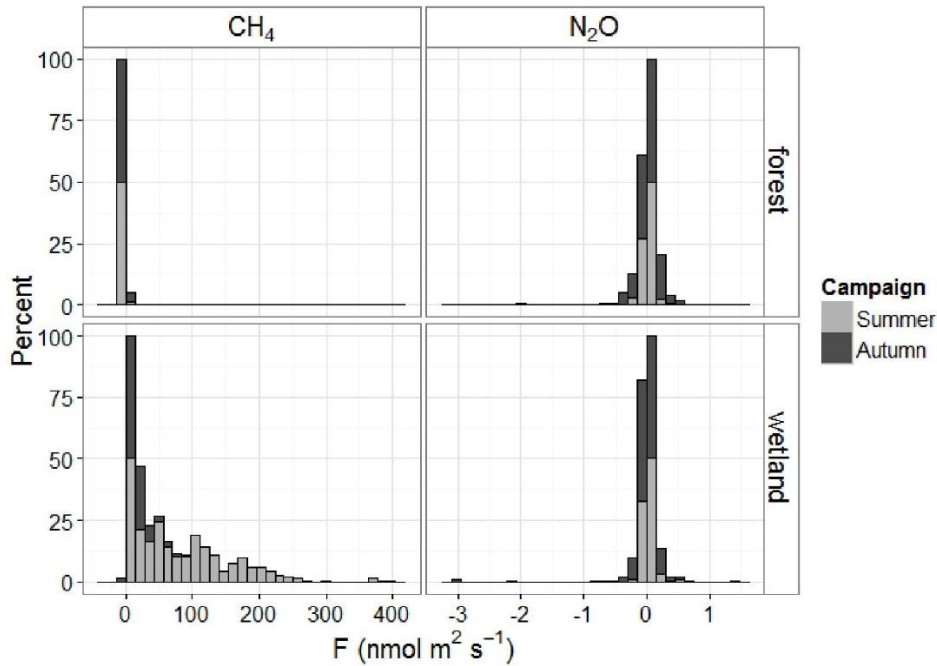

Figure 1. Frequency plot showing distribution of all fluxes across both campaign periods

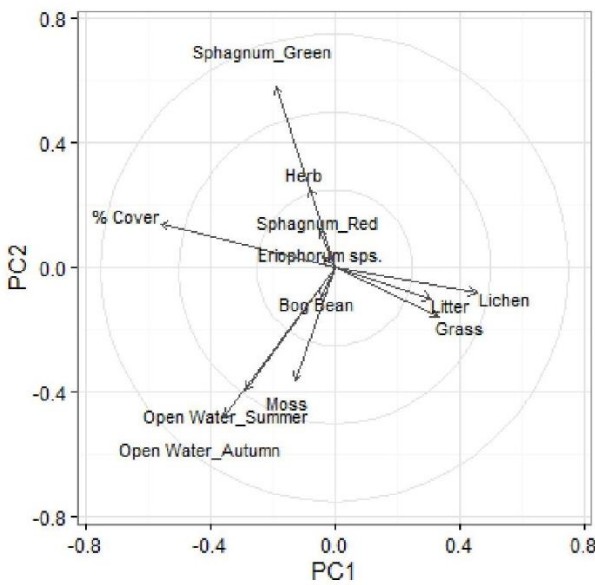

Figure 2. Loading values for principal components 1 and 2 of the chamber vegetation analysis

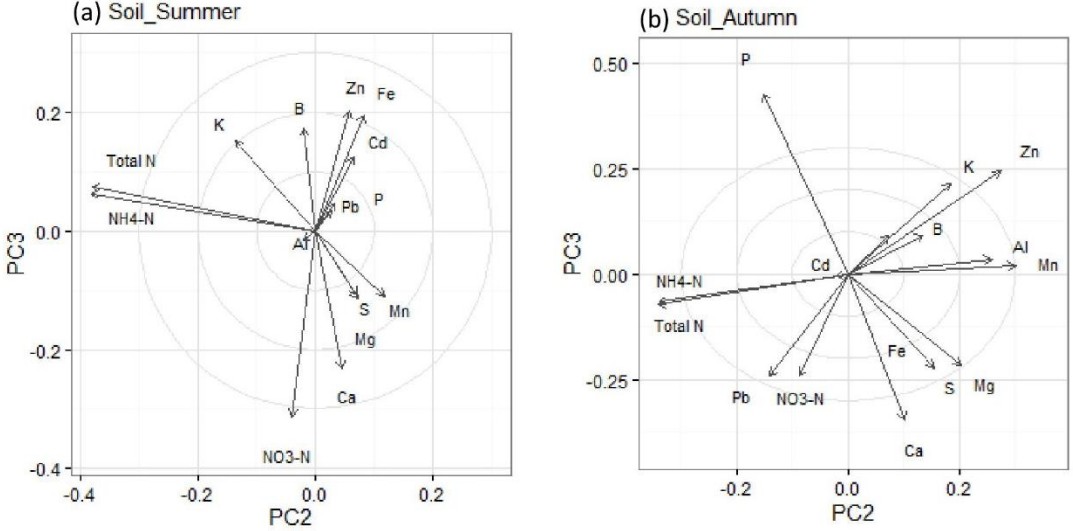

Figure 3. Loading values for principal components 2 and 3 of the chamber soil concentration analysis

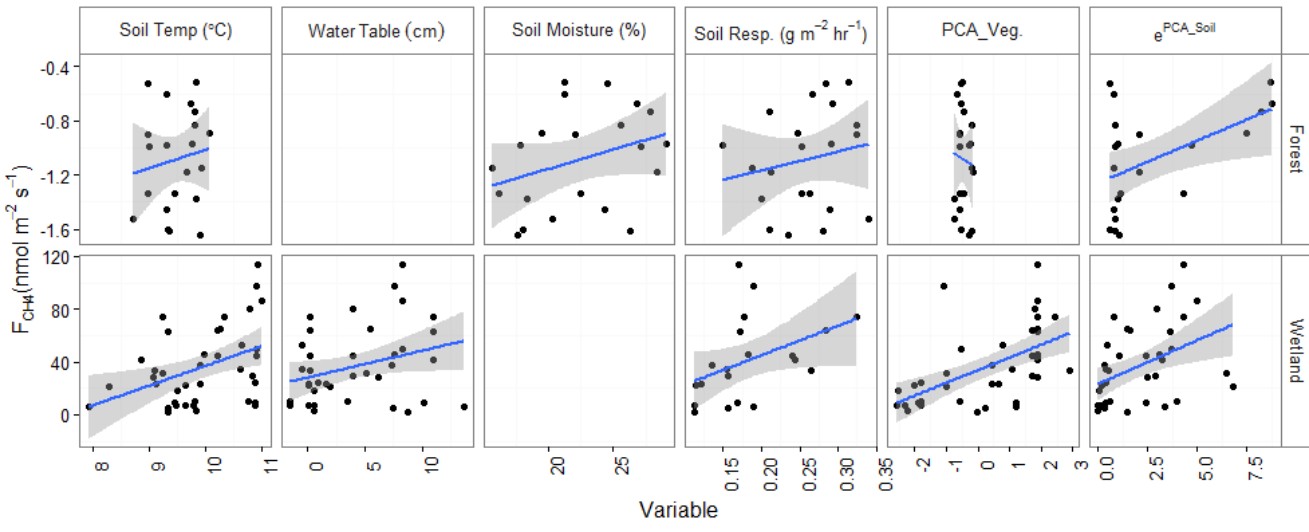

Figure 4. Relationships between geometric mean CH$_4$ flux (nmol m$^{-2}$ s$^{-1}$) against measured environmental variables across full sampling period. Text refers to the results from statistical correlations were 'ns' refers to a non-significant results, * and ** represent P < 0.05 and P < 0.01, respectively.

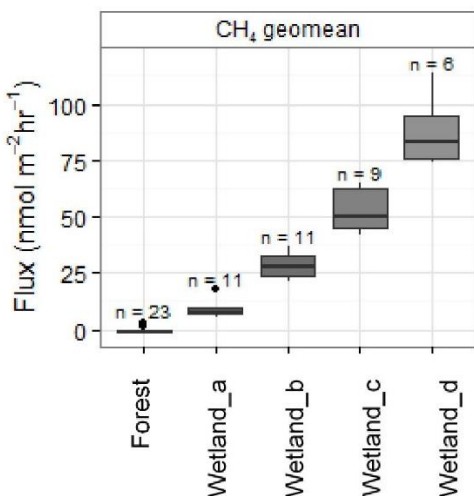

Figure 5. Chambers clustered based on emissions with n indicating the number of chambers within each group.

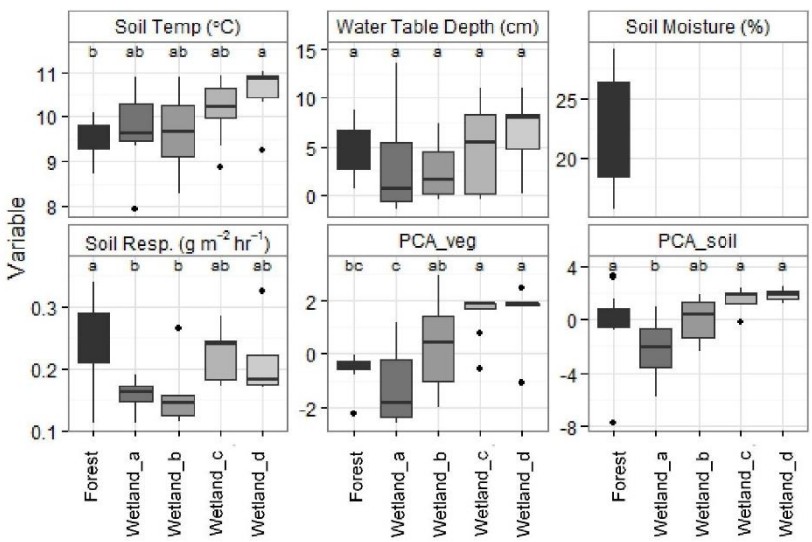

Figure 6. Boxplots showing range of measured environmental variables within each of the CH$_4$ clusters. Letters represent results from Tukeys family test statistic where clusters with similar letters are not significantly different from one another at 95% confidence level. Clusters Wetland_a to Wetland_d represent groups with sequentially increasing CH$_4$ emissions. Note water table was not measured within the forest plots therefore the water table values given in the 'forest' cluster actually represent only the 2 wetland chambers which have quantitatively been assigned to this cluster and have therefore been excluded.

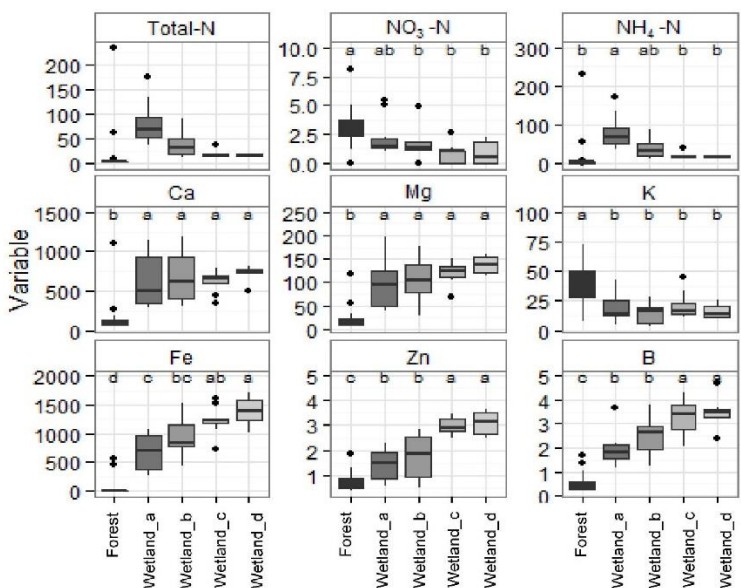

Figure 7. Boxplots showing range of soil variables within each of the $CH_4$ clusters. Units represent probe supply rate (µg per 10 cm² across burial period). Letters represent results from Tukeys family test statistic where clusters with similar letters are not significantly different from one another at 95% confidence level. Clusters Wetland_a to Wetland_d represent groups with sequentially increasing $CH_4$ emissions.

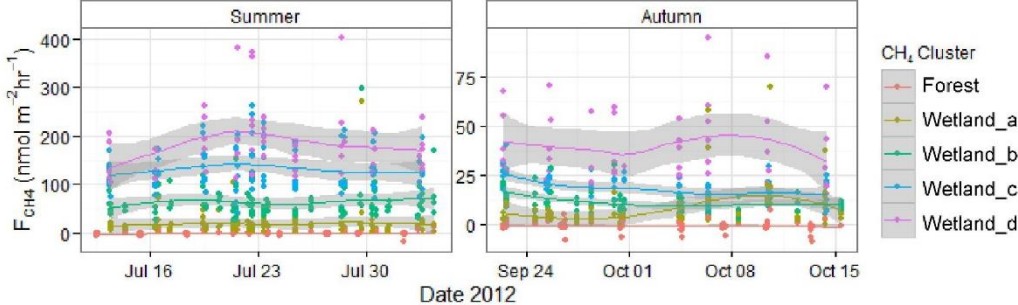

Figure 8. Temporal variability across the 2 field campaigns in $CH_4$ emissions, separated by clusters, with shaded area representing loess smoothing. Clusters Wetland_a to Wetland_d represent groups with sequentially increasing $CH_4$ emissions.

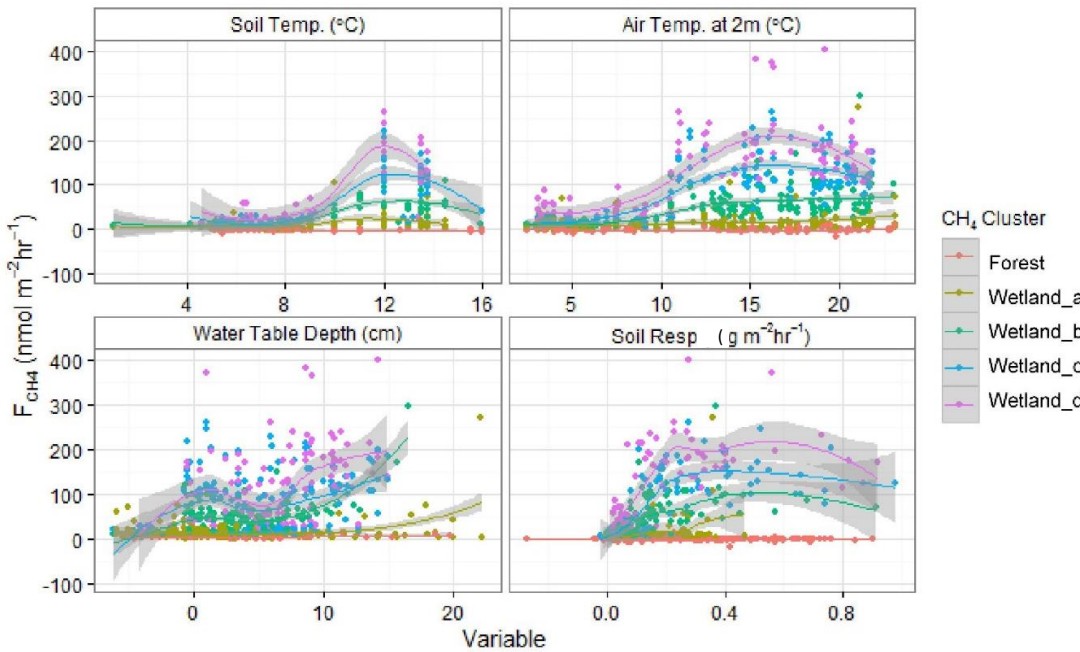

Figure 9. Drivers of temporal variability in CH$_4$ fluxes, separated by clusters, with shaded area representing loess smoothing.

Clusters Wetland_a to Wetland_d represent groups with sequentially increasing CH$_4$ emissions.

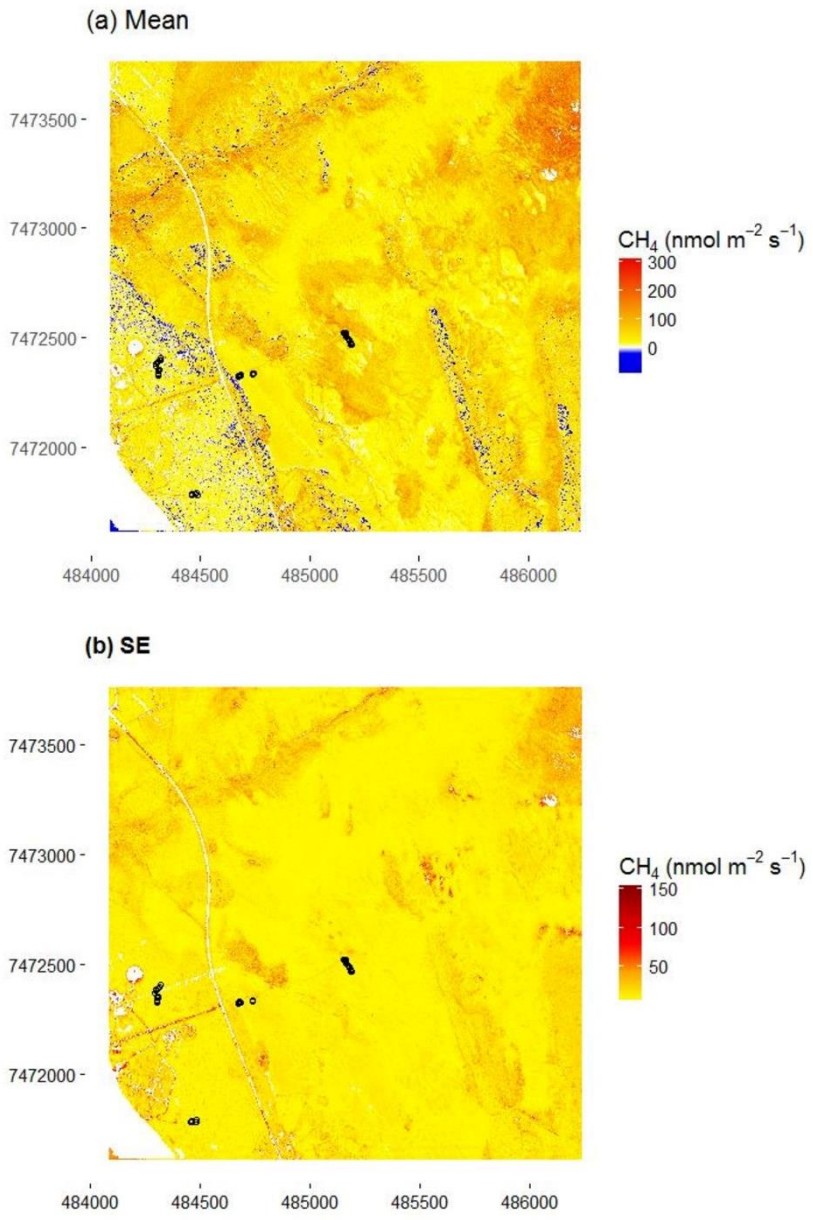

Figure 10. Mean (a) and SE (b) of CH₄ fluxes extrapolated over a 2 x 2 km area predicted from chamber flux measurements (black circles), and satellite spectral data. Coordinates are in WGS84.

