# Peer review of "Growing season CH4 and N2O fluxes from a sub-arctic landscape in northern Finland; from chamber to landscape scale"

_Biogeosciences, 2016_

## Referee Comment (RC1) · Anonymous Referee #1 · 5 Aug 2016

**GENERAL COMMENTS**

The paper you present here is a clearly written and logically constructed report on fluxes of the two important non-CO2 GHGs, CH4 and N2O, of a subarctic landscape in Northern Finland. The fluxes were measured by a static chamber technique from the main landscape elements in the region - forests and wetlands - and upscaled to an area of 4 km2 based on spectral data from a high-resolution satellite image.

The used field methods seem sound, replication of the chamber measurement is good, and the careful data analysis of the flux results is a particular strength of this study. In my opinion, some methodological issued need more detailed descriptions, and these things are specified below. The flux measurements were carried out during two relatively short campaigns in the summer and autumn season of a single year, which is

a short data collection period compared to the similar studies published during recent years. Although neither the experimental design nor the results of this study do not include any genuinely novel aspects, this kind of regional upscaling efforts are still quite rare and very much needed to improve our ability to calculate more accurate GHG balances in a large scale. This relevance of this study should be stated much better, now it is not fully convincing. In addition to this, there are several other points that require your careful consideration and before the publication of this report can be recommended.

SPECIFIC COMMENTS

The biggest problem of this manuscript is that the relevance of this particular study is not argumented well enough. This concerns both the introduction section and conclusions, and it leaves the reader with the feeling that you were not very sure of the importance of the study yourselves. More specifically: In the introduction you base the importance of the study on large SOC pool in high-latitude soils and uncertainties of the carbon-climate feedback. Since you are not measuring $CO_2$ fluxes that represent the most of the C gas fluxes between ecosystems and atmosphere, you should much more emphasize the importance of the non-$CO_2$ GHGs instead (higher radiative forcing on weight-unit-basis, uncertainties in the drivers of $CH_4$ fluxes, almost completely lacking knowledge on the distribution of $N_2O$ fluxes in high-latitude ecosystems...). The text in the abstract on lines 9-11 is a good start, but it belongs to the introduction section, since abstract should not contain ideas not mentioned in the main text of the manuscript. You should also put the $CH_4$ and $N_2O$ emissions into context, and mention clearly enough their secondary importance relative to $CO_2$.

Similarly, the discussion/conclusion section does not fully convince of the importance of the study. It is very good to point out the uncertainties of the presented results, but at the present state the conclusion chapter does not fully justify, why this study should be published as an important contribution to the field.

I find that the upscaling exercise is the most interesting part of the study, and should be

more emphasized in the paper, e.g., at the expense of the discussion on the impact of the water table level on CH4 flux that does not reach very clear conclusions. A review of similar upscaling efforts is needed. Are there many previous studies like this in the subarctic region, how about in the rest of northern Scandinavia? Are the methods used here similar or very different compared to the previous studies? What do we learn here that was not previously known?

The N2O fluxes from the studied plots were mostly not statistically different from zero. However, the results of the N2O fluxes are too much down-tuned in the manuscript text. Based on results from the last decade, there are surfaces in the subarctic and Arctic that have potential for N2O emissions (Elberling et al. 2010 NGeo, Marushchak et al. 2011 GCB, Abbott et al. 2015 GCB), although N2O is still rarely included in GHG inventories in the north. With this in view, it is important to screen various high-latitude ecosystems for N2O fluxes and also produce base-case flux balances against which possible climate change induced changes in the fluxes can be observed. The "zero-result" is not irrelevant, but it is important knowledge, which should be much stronger stated in the manuscript.

Here are some additional minor comments:

\*\*\*Abstract The abstract seems rather long to me. Could you make it more compact, concentrating just to the main outcome of the study? This would make the main message appear stronger to the reader. Page 1, line 2: Why should the ecosystems be described as consistent sinks or sources, if you can with high confidence state that the emissions are negligible? It is as important results. Now, one gets an impression that after so many flux measurements you still do not know anything about the N2O dynamics.

\*\*\*Introduction Page 2, lines 5 and 9: emissions of what? Please specify! Page 2, line 11: Here, you mention permafrost thaw as one of the secondary drivers of GHG emissions, but you do not tell in the site description if your site had permafrost or not.

Please, add this information in the site description. Page 2, lines 16-18: This is very general. How does this particular study answer to this need? What does it give that is not yet known?

***Methods Page 4, line 17: Here, you mention that the intermediate enclosure time was 15 years, while later (page 5, line 6) you say that it was 12 years. Which one is correct? Page 5, lines 7 and 8: Even if you want to avoid subjective classification of the wetland plots, and rather rely to clustering analysis, it should be easy to distinguish between ridges and flarks. Please, mention how many of your collars were located in these different mire microforms, and does this represent the proportional coverage of these microforms. This is relevant knowledge for the later upscaling exercise (upscaling based on simple averaging within wetland and forest classes). Page 5, lines 13-15: Starting the measurements so soon after the installation of the flux collars is well enough justified here, but I am missing details on how the disturbance caused by the field workers was minimized. Did you construct boardwalks in the vicinity of study plots? Did you observe (CH4) ebullition events during the measurements, and do you think they were natural or caused by people? If yes, how large proportion of the flux measurements you had to exclude for this reason? Page 5, lines 29-30: If it includes respiration from ground vegetation, ecosystem respiration would be more accurate term than soil respiration. You can anyway determine what was included (not the respiration from taller vegetation due to the methodological limitations). Page 5, line 33: 'vegetation coverage' instead of just 'vegetation' would be more precise. Page 6, line 1: Please, add a reference on PRS and/or some specification on what they sample and by which principle? Is it just collection of soil pore water, from which nutrients are analyzed or something else? A list of the measured ions would also be good to include here. Page 6, lines 8-12: Please, specify the criteria used to include or exclude the flux data for analysis, and mention (here, or in the results) how many percent of the fluxes had to be rejected. Page 6, line 24-25: Did you try the correlations on the level of single plots to investigate the drivers of temporal variability? Sometimes there can be large variability even at small scale, and this is needed to reveal the factors of

behind the variability. What made you think that the plots with similar flux magnitude would have similar mechanistic behavior? Page 7, line 3: It does not seem correct to state that the uncertainty of the N2O fluxes was large. On the contrary, based on the description of the field method and the data analysis, it is evident that the fluxes were near-zero with high confidence, just the sign of the small flux is uncertain. With increasing fluxes, also the absolute uncertainty usually increases, whereas it small for small fluxes. Please, revise the sentence to be more correct, for example: 'Due to low variability of N2O fluxes...'

***RESULTS Page 7, lines 8-9: Were these 8-9 % of the N2O fluxes that were significantly different from zero evenly distributed between study plots, or where there some that showed more often significant source or sink character? It is important to mention this, since it is well known that there is a high spatial variability of N2O emissions – where there any plots that were clearly sinks or sources? Page 8, line 3: What do you mean by soil concentration data, the concentration of nutrients in the soil pore water? Please, specify! Page 8, line 16: Do I understand this correctly, that you had higher fluxes from ridges with deep water tables than from flarks with high water tables? This is interesting. Is this a common observation from aapa mires? Page 8, line 24: Since this classification is very abstract, it would make sense to somehow relate it to wetland microforms, vegetation or similar. How were the flark and ridge collars distributed in these classes?

***DISCUSSION Page 10, line 17 onwards: The CH4 fluxes were not very well correlated with environmental factors. One explanation could be that the differences in vegetation cover were overruling the effect of other factors. The role of vegetation as a potential driver of spatio-temporal variability (vascular plant coverage/biomass/leaf area/plant number, productivity) is well acknowledged in previous literature on wetland CH4 emission. Particularly vascular plants are important due to methane transport and input of fresh carbon to the sediment. This is not adequately discussed and not very well addressed by experimental design. Please, add adequate discussion on this topic

in the discussion section. Page 11, line 29-33: These citations (Tupek, Turetsky) would need some mechanistical explanation, is this water table optimum of around 20 cm related to differences in plant productivity, i.e., a side product? What is the interpretation in the cited studies? Page 12, line 5: The spatial variability in temperatures is rather small. Do you think that this is a true temperature dependence, or is it more a result of another factor that is more important for CH4 flux, such as water table level? Page 12, lines 28-32: To make this discussion meaningful, you should mention, what where the proportions of wetlands and forests in the study by Hartley et al. vs. this study. Please, add this information!

***Figures Figure 4. Please, indicate the sampling period used for this representation – are the averages for both summer and autumn campaigns used? Figure 6. The water table of the forest plots seems too high – was it really at 5 cm below the surface and not different from wetland plots? How do you explain this?

TECHNICAL CORRECTIONS

Figure 3. In the figure caption, you mention PC 1 and 2, while PC 2 and 3 are shown in the figure. Please, check this.

---

## Referee Comment (RC2) · Anonymous Referee #2 · 12 Aug 2016

The paper "Growing season $CH_4$ and $N_2O$ fluxes from a sub-arctic landscape in northern Finland" is very well structured and is written with very good, fluent language. The study based on the state of the art methods of chamber measurements (at least for $CH_4$ and $N_2O$). The topic fits well within the scope of 'Biogeosciences'. Although the $CH_4$ and $N_2O$ measurements do not provide new insights, the subject of the study is very important, since reliable but simple upscaling approaches for GHG are still rare in literature, but are urgently needed.

I agree with the comments by the other referee but however, I have some additional remarks and a number of suggestions, which I believe will improve this manuscript once addressed and need to consider before publication.

**Major comments:**

1) I would suggest to change the title since the actual one describe insufficient the intention of the study concerning the applied modeling approach to extrapolate measured $CH_4$/$N_2O$ fluxes to landscape scale.

2) In order to receive reliable mean GHG flux rates, the amount of measurements seems rather short for me, in particular when the data set is used for model building and upscaling. For upscaling to landscape scale calculated mean flux rates or emission factors should at least represent annual values. In your study, measurements were carried out during a summer and an autumn campaign with 10 and 7 to 8 single measurements during a period of 22 or 23 days. To capture the temporal variability during these periods, conducted measurements seems sufficient. Nevertheless, for the rest of the year no additional measurements were conducted, nor any estimations or literature values were given. Form several studies published in literature, it is widely known that intra and also inter annual variability can be very high which necessitate the need for long-term studies to receive reliable mean GHG flux rates. However, the harsh environment of norther Finland makes it to some point difficult to measure around the whole year. Nevertheless, measurements during springtime would have been quite useful in regard to thawing soil conditions, which perhaps resulting in a markedly different behavior of $CH_4$ emissions. Also a rough estimation of winter time fluxes or literature values should be given. Generally, I strongly recommend that this issue should be taken up in more detail in the introduction, discussion and the conclusion of the manuscript. Please further include a sentence in the abstract that the study based just on a few single measurements during a single year.

3) Your data analysis includes an interesting approach to consider the skewness of observed $CH_4$ fluxes in the calculation of means and variations. In general the issue of skewed data and the resulting error in the calculation of means and variances of those data sets is mostly disregarded in almost all studies. Therefore, I strongly support the idea to revisit this issue. Nevertheless, there are some points which have to be described in more detail or have to be considered:
a) The geometric mean is limited by the fact that variables have to be > 0. In the presented study, $CH_4$ and $N_2O$ exchange include the release and uptake of both gases. To take this into account you calculate the geometric mean of all positive and all

negative flux rates independently and from this a frequency-weighted mean? Maybe it would be helpful to include the formula of the calculation approach.

b) In contrast to the arithmetic mean, the use of ± standard deviation or standard error is not meaningful for the geometric mean. Instead, the standard deviation should be given as multiplication or division factor (Lozán and Kausch, 2007). This has to be considered in the manuscript.

c) Why do you choose the geometric mean for the estimation of mean $CH_4/N_2O$ fluxes instead of trying to apply e.g. method of moments estimators or uniformly minimum variance unbiased estimators (for this see: Parkin et al., 1988: Evaluation of statistical estimation methods for lognormally distributed variables; Parkin et al., 1990: Calculating Confidence Intervals for the Mean of a Lognormally Distributed Variable)? Can you cite any other study who calculates a geometric mean for GHG fluxes? I suggest to recalculate the mean flux rates with both methods, presented by Parkin et al., (1988) and to compare the corresponding results with the calculated geometric mean. I think this procedure will significantly contribute to reduce the uncertainty in future investigations.

**Minor comments and suggestions:**

1) Page 2, line 30: Vegetation also exerts a direct and indirect control on $N_2O$ emission! Please complement.

2) Page 3, line 7: $N_2O$ can also be produced through abiotic processes (chemodenitrification, chemical decomposition of $NH_2OH$, surface decomposition of $NH_4NO_3$; e.g. Butterbach-Bahl, 2013: Nitrous oxide emissions from soils: how well do we understand the processes and their controls?). Change the formulation of the sentence accordingly.

3) Page 5, line 12: Please ad short information's about chamber configuration: chamber height or volume, air mixing yes or no, chamber inside thermometer yes or no, rubber lip or similar to ensure air tightness during chamber placement on in situ bases, etc…

4) Page 5, line 17: How was the chamber air collected? Did you evacuated the vials previously? How do you protect the vials for air pressure differences during air transport (e.g. Glatzel and Well, 2008: Evaluation of septum-capped vials for storage of gas samples during air transport)?

5) Page 5, line 24: In the latter manuscript, you also refer to air temperature. Please describe shortly sensor type and placement, record interval, etc. Do you measure chamber inside air temperature?

6) Page 5, line 29: I recommend the term ecosystem respiration rather than soil respiration.

7) Page 5, line 30: In my point of view, the PP-Systems SCR-1 respiration chamber (150 mm height, 100 mm diameter) seems very inappropriate for measuring ecosystem respiration (or soil respiration including ground vegetation). The dimension of the chamber is by far too small to cover the predominant vegetation at your sites

investigated. Therefore, it can be assumed that this approach significantly disturbed the plants and thus markedly change the $CO_2$ fluxes. I strongly recommend to remove all related parts in the manuscript.

8) Page 6, line 18: Did you apply any transformations (or did you remove outliers) to achieve a normal distribution in the data set (e.g. for $CH_4$ fluxes) prior to the PCA? I think that this might be necessary since PCA based on parametric Pearson correlations!

9) Page 7, line 14 and following manuscript: Did you always mean geometric mean if you write mean?

10) Page 7, line 17: Did you mean $1.06 \pm 0.44$ µg N $m^{-2}$ $hr^{-1}$ instead of $s^{-1}$? (This also relates vice versa to Table 1).

11) Page 8, line, 25: Have you tested the assumptions for linear models (e.g. normal distribution of residuals, homogeneity of variances, autocorrelation etc.)? I guess that the strong skewed dataset will partly violate the assumptions of an ANOVA? Please describe your statistical procedure in the section Data analysis. Please also describe which factors (e.g. single $CH_4$ fluxes or mean group $CH_4$ fluxes, temperatures, PCA_veg, etc.) were included as fixed effects in the ANOVA. Have you tested just one factorial or also multifactorial approaches? Did you consider temporal pseudoreplication in case of chamber specific GHG fluxes?

12) Page 9, line 13: Have you tested for non-linear relationships? In case of non-normal distribution of data, Pearson correlation coefficient (r) is perhaps not the right choice as a measure for the intensity and direction of a relationship. Maybe Spearman rank correlation coefficient is more appropriate?

13) Page 9, line 27: Please mentioned that the mean $CH_4$ flux which you use for upscaling did not represent an annual mean $CH_4$ flux rate (e.g. average $CH_4$ flux over the growing season Page 12, Line 27). Have you tried to separate between summer and autumn $CH_4$ fluxes for model building and upscaling?

14) Page 10, line 1: Is the area weighting factor 61% wetland and 32% forest?

15) Page 10, line 11 to 15: Don't be too critical with the observed close to zero net $N_2O$ fluxes and the fact that no drivers for upscaling are found. Maybe gross production of $N_2O$ occurs at your sites investigated, but in the end it is an important result that both ecosystems actual did not significantly contribute to global warming through the release of $N_2O$ emissions. However, this fragile balance can change very quickly in the course of e.g. climate warming, drainage, etc. and should therefore shortly be mentioned in the discussion and conclusion. Further, it would be fine to include also $N_2O$ fluxes as an additional Figure.

**Technical corrections:**

Please check the entire manuscript in regard to consistency of units, citations (e.g. sometimes italic formation), fonts (sometimes times new roman, sometimes other formatting).

1) Page 2, line 9: are essential -> is essential

2) Page 3, line line 6: aerobic condition -> aerobic conditions

3) Page 4, line 14: in the area our -> in the area where our ..

4) Page 5, line 2 and 3: Formatting of the date: $12^{th}$ July – $2^{nd}$ August……

5) Page 5, line 14: occasions, the short -> occasions. The short …

6) Page 5, line 15: fluxes, and -> fluxes, which

7) Page 5, line 26: 5 mm instead of 5mm (maybe you mean 5 cm for dip well instead of 5 mm?)

8) 5 line 28: located equidistance -> located at equidistance …

9) Page 6, line 2 and 3: Formatting of the date…

10) Page 6, line 30: Formatting of the date…

11) Page 7, line 9: 8 and 9% instead of 9 %

12) Page 7, line 14: both units mg C $m^{-2-}$ $hr^{-1}$ -> mg C $m^{-2}$ $hr^{-1}$

13) Page 7, line 24 and 25: $P < 0.01$ instead of $P <0.01$

14) Page 7, line 29: emissions thus -> emissions, but …

15) Page 8, line 13: emissions wert -> emissions was …

16) Page 8, line 19: correlated $CH_4$ -> correlated to $CH_4$ …

17) Page 8, line 33: Between-group differences or Between group differences; please be consistent (relates to the entire manuscript).

18) Page 9, line 23: 45%

19) Page 9, line 27: Methane can be abbreviated. This also relates to the following manuscript.

20) Page 10, line 4: -0.06 + <0.01 -> -0.06 ± <0.01

21) Page 10, line 15: Or instead of over?

22) Page 10, line 24: Turetsky et al., 2014. -> Turetsky et al., 2014).

23) Page 11, line 32: water level was -> water level were …

24) Page 12, line 3: show are -> show is …

25) Page 12, line 31: landscape scales fluxes -> landscape scale fluxes ..

26) Page 13, line 6: Hartly et al. (2015) who's study -> Hartly et al. (2015) whose study …

27) Page 13, line 22: temperature -> soil temperature

28) Page 18, Table 1: Please note that mean represent the geometric mean.

29) Page 18, Table 3: I strongly recommend the use of an adjusted r² instead of r² since r²$_{adj.}$ considered the number of predictors in the model.

30) Page 19, Figure 1: Minus sign is missing in the unit of the X-axis

31) Page 21, Figure 3 a) and b): X and Y-axis show principle components 2 and 3 instead of 1 and 2!

32) Page 22; Figure 4: Unit of soil moisture is missing!

33) Page 24, Figure 6: The Unit of soil respiration differs from Figure 4 and Figure 9 (g m$^{-2}$ hr$^{-1}$, instead of mg m$^{-2}$ hr$^{-1}$)! Did soil respiration represent $CO_2$ or $CO_2$-C? See also Minor comments and suggestions Nr. 7

34) Page 25, Figure 7: Units are missing!

---

## Author Comment (AC1) · 20 Sep 2016

**Response to reviewers comments on "Growing season $CH_4$ and $N_2O$ fluxes from a sub-arctic landscape in northern Finland" by Kerry J. Dinsmore et al.**

**Referee #1**

*GENERAL COMMENTS*

**The paper you present here is a clearly written and logically constructed report on fluxes of the two important non-$CO_2$ GHGs, $CH_4$ and $N_2O$, of a subarctic landscape in Northern Finland…The used field methods seem sound, replication of the chamber measurement is good, and the careful data analysis of the flux results is a particular strength of this study… this kind of regional upscaling efforts are still quite rare and very much needed to improve our ability to calculate more accurate GHG balances in a large scale.**

We thank reviewer 1 for their positive comments and constructive criticism, we believe the edits described below will significantly improve the original manuscript. In particular we have significantly edited the introduction and discussion to reflect the comments regarding manuscript focus and the relevance of $N_2O$ fluxes despite their small magnitude.

**The flux measurements were carried out during two relatively short campaigns in the summer and autumn season of a single year, which is a short data collection period compared to the similar studies published during recent years.**

We agree the data collection period is limited for a temporal analysis if annual or inter-annual fluxes were the focus however the aims of the study were to a) consider the drivers of $CH_4$ and $N_2O$ fluxes which given the range of variables observed we believe we can achieve well with the dataset, and b) to upscale from chamber to landscape level. These aims, particularly the upscaling, have hopefully now been brought out more with the edits suggested by both reviewers. See comments below.

*SPECIFIC COMMENTS*

**The biggest problem of this manuscript is that the relevance of this particular study is not argumented well enough. More specifically: In the introduction you base the importance of the study on large SOC pool in high-latitude soils and uncertainties of the carbon-climate feedback. Since you are not measuring $CO_2$ fluxes that represent the most of the C gas fluxes between ecosystems and atmosphere, you should much more emphasize the importance of the non-$CO_2$ GHGs instead**

Agree, the introduction has now been amended to better reflect the focus on non-$CO_2$ GHGs

**The text in the abstract on lines 9-11 is a good start, but it belongs to the introduction section, since abstract should not contain ideas not mentioned in the main text of the manuscript.**

Agree, in line with the previous comment have expanded on this in the introduction.

**You should also put the $CH_4$ and $N_2O$ emissions into context, and mention clearly enough their secondary importance relative to $CO_2$.**

This has now been incorporated into the introduction.

**Similarly, the discussion/conclusion section does not fully convince of the importance of the study. It is very good to point out the uncertainties of the presented results, but at the present state the conclusion chapter does not fully justify, why this study should be published as an important contribution to the field.**

The relevant sections have now been amended to bring out the importance of the study, linking to the additional comments added in the introduction and highlighting the strength of the upscaling rather than temporal variability.

**I find that the upscaling exercise is the most interesting part of the study, and should be more emphasized in the paper, e.g., at the expense of the discussion on the impact of the water table level on CH$_4$ flux that does not reach very clear conclusions. A review of similar upscaling efforts is needed. Are there many previous studies like this in the subarctic region, how about in the rest of northern Scandinavia? Are the methods used here similar or very different compared to the previous studies? What do we learn here that was not previously known?**

We have tried to emphasise the importance of the upscaling more throughout the paper, including more discussion on previous similar efforts, which are very few. Whilst we highlight that the correlations used here to upscale to spectral data are limited to the area from which data was collected, i.e. the formulas could not be used to upscale across the wider northern Scandinavian area, the method itself worked well and could be applied to similar small scale chamber studies to improve estimates over their specific landscapes. Whilst enhancing this scaling discussion we have chosen not to replace the water table discussion. Whilst we agree patterns were not clear, what we did find was counter to much previous literature making it in itself an important finding, even if further work is required to narrow down the control mechanisms.

**The N$_2$O fluxes from the studied plots were mostly not statistically different from zero. However, the results of the N$_2$O fluxes are too much down-tuned in the manuscript text. Based on results from the last decade, there are surfaces in the subarctic and Arctic that have potential for N$_2$O emissions (Elberling et al. 2010 NGeo, Marushchak et al. 2011 GCB, Abbott et al. 2015 GCB), although N$_2$O is still rarely included in GHG ecosystems for N$_2$O fluxes and also produce base-case flux balances against which possible climate change induced changes in the fluxes can be observed. The "zeroresult" is not irrelevant, but it is important knowledge, which should be much stronger stated in the manuscript.**

We agree with the reviewer that a zero result is still an important one however our data did not lend itself to an analysis of drivers of N$_2$O and sufficient correlations were not present to enable an upscaling, hence the inevitable omission from much of the discussion. We do however acknowledge that the magnitude of the flux, which as rightly pointed out is an important baseline for future studies, has become lost within the manuscript. Where possible within the discussion and especially conclusions we have tried to emphasis the result more and based on previous comments have added more on N$_2$O to the introduction, utilising the helpful references the referee has suggested.

**ABSTRACT**

**The abstract seems rather long to me. Could you make it more compact, concentrating just to the main outcome of the study?**

We have reduced the length of the abstract as suggested, retaining the results but reducing the interpretation, e.g. paragraph 3 is much more succinct, now reading

*'We found a weak negative relationship between CH$_4$ emissions and water table depth in the wetland, with emissions decreasing as the water table approached and flooded the soil surface. Temperature was also an important driver of CH$_4$ with emissions increasing to a peak at approximately 12°C. Little could be determined about the drivers of N$_2$O emissions given the small magnitude of the fluxes.'*

**Page 1, line 2: Why should the ecosystems be described as consistent sinks or sources, if you can with high confidence state that the emissions are negligible?**

We have removed the reference to N$_2$O sources or sinks in the abstract and now state only that N$_2$O results were near-zero across both ecosystems.

**INTRODUCTION**

**Page 2, lines 5 and 9: emissions of what? Please specify!**

This has been amended to read *'GHG emissions are still poorly constrained (e.g. Bridgham et al., 2013)'*

**Page 2, line 11: Here, you mention permafrost thaw as one of the secondary drivers of GHG emissions, but you do not tell in the site description if your site had permafrost or not.**

The site did not contain permafrost and this has now been added to the site description as suggested.

**Page 2, lines 16-18: This is very general. How does this particular study answer to this need? What does it give that is not yet known?**

We look at the drivers of $CH_4$ in significant detail, in particular the water table and temperature relationships and also the use of the soil probes to consider nutrient availability. We believe this is well discussed and highlighted throughout the results section. However in light of the referee's comments we've added to the discussion the importance of the water table result, in particular that it is different to what much of the previous literature states, and what this tells us about the underlying mechanisms.

**Methods Page 4, line 17: Here, you mention that the intermediate enclosure time was 15 years, while later (page 5, line 6) you say that it was 12 years**

This has been corrected to 12 years.

**Page 5, lines 7 and 8: Even if you want to avoid subjective classification of the wetland plots, and rather rely to clustering analysis, it should be easy to distinguish between ridges and flarks. Please, mention how many of your collars were located in these different mire microforms, and does this represent the proportional coverage of these microforms. This is relevant knowledge for the later upscaling exercise (upscaling based on simple averaging within wetland and forest classes).**

The chambers were located across the range of water levels however clear ridges and flarks were not easily distinguished at that time of year; whilst hummocks and hollows were visible as microtopographical features of the wetland these are smaller than the resolution of the satellite are therefore not a useful classification in this context. The chambers covered a range of water levels and a range of vegetation types, some becoming submerged for extended periods but all dry at some point within the 2 sampling campaigns. We have therefore chosen not to subjectively label each chamber.

**Page 5, lines 13-15: … I am missing details on how the disturbance caused by the field workers was minimized. Did you construct boardwalks in the vicinity of study plots? Did you observe ($CH_4$) ebullition events during the measurements, and do you think they were natural or caused by people? If yes, how large proportion of the flux measurements you had to exclude for this reason?**

The following text has now been added to the methods. Sampling was carried out from existing boardwalks therefore human-induced ebullition was not a problem and no fluxes were omitted due to this.

*'Wetland chambers were located so that sampling could be carried out from an existing boardwalk, this served the dual purpose of avoiding disturbance during chamber enclosure and minimised the environmental impact of footfall on the site. The ground surface within the forest plots was considered to be solid and therefore no such precautions were required.'*

**Page 5, lines 29-30: If it includes respiration from ground vegetation, ecosystem respiration would be more accurate term than soil respiration. You can anyway determine what was included (not the respiration from taller vegetation due to the methodological limitations).**

We have chosen to keep the definition of soil respiration as ecosystem respiration implies much more vegetation than was included, we have edited the appropriate description as below to clarify.

*'Soil respiration (note whilst we refer to this as soil respiration throughout, it also includes respiration from the ground surface vegetation defined as anything with a height of less than 2 cm above ground surface), was measured using a PP-Systems SCR-1 respiration chamber'*

**Page 5, line 33: 'vegetation coverage' instead of just 'vegetation' would be more precise.**

Agree, amendment made as suggested

**Page 6, line 1: Please, add a reference on PRS and/or some specification on what they sample and by which principle? Is it just collection of soil pore water, from which nutrients are analyzed or something else? A list of the measured ions would also be good to include here.**

The following text has been added as requested:

'*The PRS probes utilise ion-exchange resin membranes to provide an index of relative plant nutrient availability (Hangs et al., 2002), measured ions included total N, NO3-N, NH4-N, Ca, Mg, K, P, Fe, Mn, Zn, B, S, Pb, Al, and Cd.* '

**Page 6, lines 8-12: Please, specify the criteria used to include or exclude the flux data for analysis, and mention (here, or in the results) how many percent of the fluxes had to be rejected.**

No data was excluded from the analysis, we rely on the GCFlux model to accurately choose the best fit method to determine the fluxes with 4 sampling points within each chamber deemed sufficient to prevent a single point overly influencing the final calculated flux. An uncertainty is calculated for each flux during the GCFlux processing and this has been used to say whether the data gives confidence in the calculated flux, i.e. leading to the discussion about fluxes not being significantly different to zero.

**Page 6, line 24-25: Did you try the correlations on the level of single plots to investigate the drivers of temporal variability? Sometimes there can be large variability even at small scale, and this is needed to reveal the factors behind the variability. What made you think that the plots with similar flux magnitude would have similar mechanistic behavior?**

Yes we considered individual fluxes as suggested by the reviewer however the high variability within each chamber produced messy results which did not show clear or useful conclusions. Grouping the data produced a much clearer picture that could be analysed. We explored multiple options for grouping the data including based on vegetation alone and on soil and other environmental factors. Using flux magnitude to group the chambers ensured that those variables that were important in controlling the spatial variability were included as part of the analysis e.g. the proportion of *Sphagnum* within the chambers was captured as a by-product of this grouping approach. Therefore whilst the group is based on flux magnitude we do not assume that this itself is related to mechanistic behaviour, rather it provides a method of capturing those variables that do.

**Page 7, line 3: It does not seem correct to state that the uncertainty of the $N_2O$ fluxes was large.**

What was meant was the variability 'relative' to the flux was large. This has now been amended and clarified in the text.

**RESULTS**

**Page 7, lines 8-9: Were these 8-9 % of the $N_2O$ fluxes that were significantly different from zero evenly distributed between study plots.**

Yes these were evenly distributed, no patterns could be seen with particular groups showing significant sources or sinks. The following text has been added to clarify:

"*The proportion of chambers displaying significant $N_2O$ fluxes could not be linked to any measured environmental factors and were distributed randomly across the dataset*".

**Page 8, line 3: What do you mean by soil concentration data...**

This has been amended to read "*to summarise the available soil nutrient availability data from the PRS probes*"

**Page 8, line 16: Do I understand this correctly, that you had higher fluxes from ridges with deep water tables than from flarks with high water tables? This is interesting. Is this a common observation from aapa mires?**

We found highest emissions in groups containing low proportions of open water (open water being a feature of flarks) and high proportions of *Sphagnum*. This finding is discussed in detail within the discussion, pg 11, with references such as Pelletier et al. (2007) describing similar water table dynamics. Much of the literature shows higher $CH_4$ emissions from flarks than ridges. However in our case we are not measuring from true flarks where the water level is above the soil surface almost permanently and vegetation is no longer present. Here we experienced fluctuating water levels with vegetated chambers becoming submerged therefore the production and consumption mechanisms, and importantly the soil redox potentials are likely to be different to those the reviewer is referring to. We have added to the text, particularly the field description, to clarify.

**Page 8, line 24: Since this classification is very abstract, it would make sense to somehow relate it to wetland microforms, vegetation or similar. How were the flark and ridge collars distributed in these classes?**

Whilst initial chamber placement used prior expertise of the likely variability due to vegetation and microtopography, we have purposely kept the cluster analysis quantitative. We have now added more detail on the microtopography within the discussion to hopefully address the reviewers comments.

*DISCUSSION*

**Page 10, line 17 onwards: The CH$_4$ fluxes were not very well correlated with environmental factors. One explanation could be that the differences in vegetation cover were overruling the effect of other factors… Please add adequate discussion on this topic in the discussion section.**

We acknowledge the importance of vegetation cover and in fact found vegetation to be the primary correlate with CH$_4$ emissions. Vegetation data comprised % coverage values which themselves were not normally distributed and could not therefore be individually tested against chamber emissions using standard statistical methods. We therefore chose to summarise the variability in chamber specific vegetation cover using a principal components approach, the resulting PCA scores were then used in further statistical analysis. From this we found a positive significant relationship between the PC2 value and CH$_4$ emissions and go on to explain that PC2 relates primarily to *Sphagnum* cover (4.1 Drivers of CH$_4$ emissions). We have highlighted the importance of vegetation cover in predicting long term antecedent water table conditions but did not discuss the further mechanistic reasons linking CH$_4$ and vegetation functional group. We thank the reviewer for highlighting this, the section has now been amended with further discussion added as requested.

**Page 11, line 29-33: These citations (Tupek, Turetsky) would need some mechanical explanation, is this water table optimum of around 20 cm related to differences in plant productivity, i.e., a side product?**

Agree, this was not well addressed in the submitted manuscript. We have since added the following paragraph.

*"Potential explanations for the inhibition of CH$_4$ emissions at high water levels given by Turetsky et al. (2014) include limited diffusion of CH$_4$ through standing water as discussed above, reduced CH$_4$ production due to lower plant biomass and associated labile C inputs, or unfavourable redox conditions resulting from inputs of oxygen rich water potentially containing alternative electron acceptors. Whilst we saw no clear correlations between the percentage of bare soil and that of open water in our chambers, a reduction in plant activity may have occurred during submersion so reduction in C inputs for methanogenesis cannot be ruled out. Neither do we have the data to rule out a change in redox potential due to water flow. A more detailed analysis under controlled conditions would be required to accurately explain the mechanism for high water CH$_4$ limitation at this site."*

**Page 12, line 5: The spatial variability in temperatures is rather small. Do you think that this is a true temperature dependence, or is it more a result of another factor that is more important for CH$_4$ flux, such as water table level?**

We believe this comment is already addressed as stated below (Pg 12 ln 8)

*"The spatial variability in soil temperature is likely to be linked to a combination of soil water content and the surface reflectance of the vegetation cover"*

**Page 12, lines 28-32: To make this discussion meaningful, you should mention, what where the proportions of wetlands and forests in the study by Hartley et al. vs. this study. Please, add this information!**

Agree, this information has now been included alongside the relevant discussion.

*"Whereas we carried out our upscaling over an area characterised by 61% wetlands and 32% forest, the landscape unit measured by Hartley et al. (2015) contained only ~22% wetland (classified as both mire and mire edge) and 60% forest."*

*FIGURES*

**Figure 4. Please, indicate the sampling period used for this representation – are the averages for both summer and autumn campaigns used?**

Data was used from full sampling period. This information has been added to the figure legend to clarify

**Figure 6. The water table of the forest plots seems too high – was it really at 5 cm below the surface and not different from wetland plots? How do you explain this?**

As stated on page 8, ln 22, "*the cluster identified with the lowest emissions contained all the forest chambers and an additional two low emitting wetland chambers*". It is these wetland chambers which have skewed the water table data in the figure. Water table was not measured in the forest plots as soil moisture was deemed a more appropriate measure of soil water conditions. This clearly leads to a false impression in the figure, we have therefore included a note in the legend to explain this detail.

*TECHNICAL CORRECTIONS*

**Figure 3. In the figure caption, you mention PC 1 and 2, while PC 2 and 3 are shown in the figure. Please, check this.**

This has now been corrected.
* * *
**Referee #2**

**The paper "Growing season CH$_4$ and N$_2$O fluxes from a sub-arctic landscape in northern Finland" is very well structured and is written with very good, fluent language. The study based on the state of the art methods of chamber measurements (at least for CH$_4$ andN$_2$O). The topic fits well within the scope of 'Biogeosciences'. Although the CH$_4$ and N$_2$O measurements do not provide new insights, the subject of the study is very important, since reliable but simple upscaling approaches for GHG are still rare in literature, but are urgently needed.**

We thank the referee for their positive comments and as with referee 1, we feel the edits made in response to their constructive criticism has significantly improved the manuscript.

*MAJOR COMMENTS:*

**1) I would suggest to change the title since the actual one describe insufficient the intention of the study concerning the applied modelling approach to extrapolate measured CH$_4$/N$_2$O fluxes to landscape scale.**

Agree, the tile has now been changed to '*Growing season CH$_4$ and N$_2$O fluxes from a sub-arctic landscape in northern Finland; from chamber to landscape scale'*.

2) **In order to receive reliable mean GHG flux rates, the amount of measurements seems rather short for me… For upscaling to landscape scale calculated mean flux rates or emission factors should at least represent annual values…. also inter annual variability can be very high which necessitate the need for long-term studies to receive reliable mean GHG flux rates. …measurements during springtime would have been quite useful in regard to thawing soil conditions, which perhaps resulting in a markedly different behaviour of CH$_4$ emissions…. a rough estimation of winter time fluxes or literature values should be given. Generally, I strongly recommend that this issue should be taken up in more detail in the introduction, discussion and the conclusion of the manuscript. Please further include a sentence in the abstract that the study based just on a few single measurements during a single year.**

We acknowledge the limitations pointed out by the referee and have, as advised, addressed these issues within the relevant sections of the introduction, discussion and conclusions. In particular we have used longer term studies such as Jackowicz-Korczyński *et al.* (2010) who found 65% of CH$_4$ emissions occurred during the summer, 25% during shoulder seasons and only 10% during winter, to put our results in context. We have also edited the abstract as advised to now read: '*Hence this study aims to increase our understanding of what drives fluxes of CH$_4$ and N$_2$O in a subarctic forest/wetland landscape during peak summer conditions and into the shoulder season,…*'

*Jackowicz-Korczyński, M., T. R. Christensen, K. Bäckstrand, P. Crill, T. Friborg, M. Mastepanov, and L. Ström (2010), Annual cycle of methane emission from a subarctic peatland, J. Geophys. Res., 115, G02009, doi:10.1029/2008JG000913.*

**3) Your data analysis includes an interesting approach to consider the skewness of observed CH₄ fluxes in the calculation of means and variations. In general the issue of skewed data and the resulting error in the calculation of means and variances of those data sets is mostly disregarded in almost all studies…**

**a) The geometric mean is limited by the fact that variables have to be > 0. In the presented study, CH₄ and N₂O exchange include the release and uptake of both gases. To take this into account you calculate the geometric mean of all positive and all negative flux rates independently and from this a frequency-weighted mean? Maybe it would be helpful to include the formula of the calculation approach.**

The formula for the calculation of the geomean from temporal fluxes is set out below, where $\bar{F}_{geom}$ is the geometric mean flux across the time period, $P_E$ and $P_U$ are the proportion of individual fluxes which represent emissions and uptake, respectively, n is the number of fluxes in the appropriate category, and E and U represent individual emission and uptake vales, respectively.

$$\bar{F}_{geom} = P_E \sqrt[n]{E_1.E_2\ldots E_n} - P_U \sqrt[n]{U_1.U_2\ldots U_n}$$

However, as the geometric mean itself is a well-defined parameter, we feel the description already included in the methods (see below), more simply represents the calculation. For now we have left the manuscript as is however if advised we are happy to edit to include the above formula.

*'Where periods of uptake and emission were both present within a time series, geometric means were calculated for each flux direction independently. The presented geometric means are the frequency-weighted sum of emissions and uptake'.*

**b) In contrast to the arithmetic mean, the use of ± standard deviation or standard error is not meaningful for the geometric mean. Instead, the standard deviation should be given as multiplication or division factor (Lozán and Kausch, 2007). This has to be considered in the manuscript.**

As geometric means are used as a first step to summarise temporal data and are not presented in their own right, standard deviations are only given when arithmetic means are calculated. We have edited the text where appropriate to ensure the reader is not confused as to which mean is being presented.

**c) Why do you choose the geometric mean for the estimation of mean CH₄/N₂O fluxes instead of trying to apply e.g. method of moments estimators or uniformly minimum variance unbiased estimators (for this see: Parkin et al., 1988: Evaluation of statistical estimation methods for lognormally distributed variables; Parkin et al., 1990: Calculating Confidence Intervals for the Mean of a Lognormally Distributed Variable)? Can you cite any other study who calculates a geometric mean for GHG fluxes? I suggest to recalculate the mean flux rates with both methods, presented by Parkin et al., (1988) and to compare the corresponding results with the calculated geometric mean. I think this procedure will significantly contribute to reduce the uncertainty in future investigations.**

The geometric mean is a standard mathematical descriptor that avoids bias due to extreme measurements in skewed datasets. When summarising the temporal dataset, if a straight arithmetic mean was used, as is often the case, the assumption is that essentially a straight line can be applied between time points, however, as we know from previous literature that the recorded 'spikes' in the dataset are likely to last a lot shorter time period than that between our measurements, this gives an unrealistically high estimate to be used in further calculations. In this instance when prior knowledge of normal temporal variability exists the geometric mean is a more logical approach. The primary issue which prevents its common use is its inability to deal with negative vales. As we can separate our datasets into emissions and uptakes this problem is easily overcome. Other published studies have also presented geometric means to summarise GHG data e.g. Cowan et al Biogeosciences 12, 1585-1596, Dinsmore et al Soil Biology and Biochemistry, 41 (6). 1315-1323. 10.1016/j.soilbio.2009.03.022.

We acknowledge that this is an area that would benefit from a full statistical analysis and comparison of methods as pointed out by the referee. However without more measurements, e.g. a high frequency time series, where actual population means and variances are known to a high degree of certainty, we cannot carry out a proper comparison of the methods listed above, we would simply obtain a variety of estimated means without knowing which was most appropriate. This is something that would, and we believe should, be the focus of another study. In this instance we have chosen to keep our method as is, as it is the simplest of the options presented with no clear disadvantages and

an amendment would require all the analysis, figures and tables to be redone. We are however happy to reconsider at the request of the editor.

*MINOR COMMENTS AND SUGGESTIONS*:

**1) Page 2, line 30: Vegetation also exerts a direct and indirect control on $N_2O$ emission!**

A reference to plant-mediated transport has now been added to the following paragraph which discusses $N_2O$ emissions.

**2) Page 3, line 7: $N_2O$ can also be produced through abiotic processes (chemodenitrification, chemical decomposition of NH2OH, surface decomposition of NH4NO3; e.g. Butterbach-Bahl, 2013: Nitrous oxide emissions from soils: how well do we understand the processes and their controls?). Change the formulation of the sentence accordingly.**

The sentence has been deleted. We focus on measuring the drivers, and not the processes involved in the N2O production/emission. In hind side it would be better to just refer to nitrification processes and denitrification processes, which was done in the preceding sentence.

**3) Page 5, line 12: Please ad short information's about chamber configuration: chamber height or volume, air mixing yes or no, chamber inside thermometer yes or no, rubber lip or similar to ensure air tightness during chamber placement on in situ bases, etc…**

These details have been added as requested

**4) Page 5, line 17: How was the chamber air collected? Did you evacuated the vials previously? How do you protect the vials for air pressure differences during air transport (e.g. Glatzel and Well, 2008: Evaluation of septum-capped vials for storage of gas samples during air transport)?**

To avoid any of these problems we did not evacuate vials, instead a 100 ml air sample was withdrawn from the chamber and flushed through a 20 ml glass vial using a double needle system. This information is already in the text, page 5 line 18

**5) Page 5, line 24: In the latter manuscript, you also refer to air temperature. Please describe shortly sensor type and placement, record interval, etc. Do you measure chamber inside air temperature?**

Air temperature was obtained from a met station on site, details have now been added.

**6) Page 5, line 29: I recommend the term ecosystem respiration rather than soil respiration.**

See comment above to reviewer #1

**7) Page 5, line 30: In my point of view, the PP-Systems SCR-1 respiration chamber (150 mm height, 100 mm diameter) seems very inappropriate for measuring ecosystem respiration (or soil respiration including ground vegetation). The dimension of the chamber is by far too small to cover the predominant vegetation at your sites investigated. Therefore, it can be assumed that this approach significantly disturbed the plants and thus markedly change the $CO_2$ fluxes. I strongly recommend to remove all related parts in the manuscript.**

The PP-Systems SCR-1 respiration chamber is a well-used method with data from it published many times. We accept this is not a measure of ecosystem respiration which is why we have chosen to use the term soil respiration. Neither do we propose it covers all ground vegetation, this has been amended following reviewer #1's comments. We do not suggest that these represent true ecosystem $CO_2$ fluxes however they are a useful indicator of soil respiration, and therefore general conditions within the soil, so we have kept the measurements as part of the analysis. Vegetation removal to get a true soil respiration value would have caused significant disturbance so chambers were placed in an appropriate area with no or as little as possible natural vegetation coverage, text has been amended within the method section to clarify this.

**8) Page 6, line 18: Did you apply any transformations (or did you remove outliers) to achieve a normal distribution in the data set (e.g. for $CH_4$ fluxes) prior to the PCA? I think that this might be necessary since PCA based on parametric Pearson correlations!**

The skewness was primarily in the temporal dataset, as this analysis was carried out on the geometric means summarising this temporal data, the data were sufficiently much less skewed. Where non normal distributions were still a problem, log transformations were carried out. More details on this have been added in the data analysis section.

**9) Page 7, line 14 and following manuscript: Did you always mean geometric mean if you write mean?**

No, geometric means are only used to summarise the highly skewed temporal datasets, arithmetic means were appropriate when considering spatial variability. This has now been clarified in the data analysis section.

**10) Page 7, line 17: Did you mean 1.06 ± 0.44 µg N m−2 hr−1 instead of s−1? (This also relates vice versa to Table 1).**

This has now been corrected, the correct unit is hr-1.

**11) Page 8, line, 25: Have you tested the assumptions for linear models (e.g. normal distribution of residuals, homogeneity of variances, autocorrelation etc.)? I guess that the strong skewed dataset will partly violate the assumptions of an ANOVA? Please describe your statistical procedure in the section Data analysis. Please also describe which factors (e.g. single $CH_4$ fluxes or mean group $CH_4$ fluxes, temperatures, PCA_veg, etc.) were included as fixed effects in the ANOVA. Have you tested just one factorial or also multifactorial approaches? Did you consider temporal pseudoreplication in case of chamber specific GHG fluxes?**

More information has now been supplied within the data analysis section including the additions below. Pseudoreplication due to temporal datasets was not an issue as only the spatial datasets were used i.e. PCA results and means.

*'In all further analysis, log transformations were applied where data-sets displayed non-normal distributions; given the time between measurements, autocorrelation within datasets was never significant'*

*'ANOVA and Tukey's pairwise comparisons were used to explore the differences in environmental variables between clusters, tested variables included means of soil temperature, water table depth and soil respiration alongside vegetation principal component and soil principal component.'*

**12) Page 9, line 13: Have you tested for non-linear relationships? In case of non-normal distribution of data, Pearson correlation coefficient (r) is perhaps not the right choice as a measure for the intensity and direction of a relationship. Maybe Spearman rank correlation coefficient is more appropriate?**

This discussion refers to the temporal dataset, relationships were non-linear but could not be modelled with simple non-linear approaches. It was not deemed necessary within the context of the manuscript to delve further into complex non-linear modelling approaches. These relationships are not described using statistics due to the complex patterns that would be oversimplified and liable to misinterpretation with summary descriptors of intensity and direction.

**13) Page 9, line 27: Please mentioned that the mean $CH_4$ flux which you use for upscaling did not represent an annual mean $CH_4$ flux rate (e.g. average $CH_4$ flux over the growing season Page 12, Line 27). Have you tried to separate between summer and autumn $CH_4$ fluxes for model building and upscaling?**

We have now added text both here and throughout the manuscript to highlight our upscaling is only valid over our sampling period between 12th July and 14th October. Whilst we set up the field campaigns with the intention to cover mid-summer and shoulder seasons, for the upscaling we do not have a long enough time series to clearly define these seasons based on fluxes or meteorological data in a way that is scientifically useful. We have therefore chosen to combine the campaigns to give us the best estimate of growing season fluxes.

**14) Page 10, line 1: Is the area weighting factor 61% wetland and 32% forest?**

This is correct, this information is already in the site description but has now been added to Pg 10 ln 1 as well.

**15) Page 10, line 11 to 15: Don't be too critical with the observed close to zero net $N_2O$ fluxes and the fact that no drivers for upscaling are found. Maybe gross production of $N_2O$ occurs at your sites investigated, but in the end it is an important result that both ecosystems actual did not significantly contribute to global warming through the release of $N_2O$ emissions. However, this fragile balance can change very quickly in the course of e.g. climate warming, drainage, etc. and should therefore shortly be mentioned in the discussion and conclusion. Further, it would be fine to include also $N_2O$ fluxes as an additional Figure.**

This has been primarily dealt with in response to reviewer 1's comments as described above and should now satisfy referee 2 also. The $N_2O$ figure was removed after significant discussion among co-authors as it was felt that it did not give the reader any useful information and the manuscript already contain a significant number of plots. We are happy to reconsider if it is felt by the editor it would be a useful addition.

**Technical corrections:**

**1) Page 2, line 9: are essential -> is essential**

Corrected

**2) Page 3, line line 6: aerobic condition -> aerobic conditions**

Corrected

**3) Page 4, line 14: in the area our -> in the area where our ..**

Corrected

**4) Page 5, line 2 and 3: Formatting of the date: 12th July – 2nd August……**

Please advise further?

**5) Page 5, line 14: occasions, the short -> occasions. The short …**

Corrected

**6) Page 5, line 15: fluxes, and -> fluxes, which**

Corrected

**7) Page 5, line 26: 5 mm instead of 5mm (maybe you mean 5 cm for dip well instead of 5 mm?)**

Changed to 5 cm

**8) 5 line 28: located equidistance -> located at equidistance …**

Kept as original wording

**9) Page 6, line 2 and 3: Formatting of the date…**

Please advise further?

**10) Page 6, line 30: Formatting of the date…**

Please advise further?

**11) Page 7, line 9: 8 and 9% instead of 9 %**

Kept as original, please advise if this is incorrect

**12) Page 7, line 14: both units mg C m-2- hr-1 -> mg C m-2 hr-1**

Corrected

**13) Page 7, line 24 and 25: P < 0.01 instead of P <0.01**

Corrected here and additional 3 instances throughout

**14) Page 7, line 29: emissions thus -> emissions, but …**

Changed to emissions, thus…

**15) Page 8, line 13: emissions wert -> emissions was …**

Kept as emissions were

**16) Page 8, line 19: correlated $CH_4$ -> correlated to $CH_4$ …**

Corrected

**17) Page 8, line 33: Between-group differences or Between group differences; please be consistent (relates to the entire manuscript).**

Changed to 'Between-group' throughout

**18) Page 9, line 23: 45%**

*Kept as original, please advise if this is incorrect*

**19) Page 9, line 27: Methane can be abbreviated. This also relates to the following manuscript.**

*Edited as suggested throughout*

**20) Page 10, line 4: -0.06 + <0.01 -> -0.06 ± <0.01**

*Corrected*

**21) Page 10, line 15: Or instead of over?**

*Word 'over' removed, now reads 'N$_2$O emissions within our landscape'*

**22) Page 10, line 24: Turetsky et al., 2014. -> Turetsky et al., 2014).**

*Corrected*

**23) Page 11, line 32: water level was -> water level were …**

*Kept as original*

**24) Page 12, line 3: show are -> show is …**

*Corrected*

**25) Page 12, line 31: landscape scales fluxes -> landscape scale fluxes ..**

*Corrected*

**26) Page 13, line 6: Hartly et al. (2015) who's study -> Hartly et al. (2015) whose study…**

*Corrected*

**27) Page 13, line 22: temperature -> soil temperature**

*Corrected*

**28) Page 18, Table 1: Please note that mean represent the geometric mean.**

*These represent arithmetic means, geometric means are only used to summarise the temporal datasets*

**29) Page 18, Table 3: I strongly recommend the use of an adjusted r² instead of r² since r²adj. considered the number of predictors in the model.**

*Adjusted r2 are used, this has been amended in table*

**30) Page 19, Figure 1: Minus sign is missing in the unit of the X-axis**

*Unsure what this refers to, minus sign is already visible, figure left as is.*

**31) Page 21, Figure 3 a) and b): X and Y-axis show principle components 2 and 3 instead of 1 and 2!**

*This has been edited in the legend.*

**32) Page 22; Figure 4: Unit of soil moisture is missing!**

*Corrected*

**33) Page 24, Figure 6: The Unit of soil respiration differs from Figure 4 and Figure 9 (g m-2 hr-1, instead of mg m-2 hr-1)! Did soil respiration represent CO$_2$ or CO$_2$-C? See also Minor comments and suggestions Nr. 7**

*Corrected*

**34) Page 25, Figure 7: Units are missing!**

*Corrected*

---

## Author Response (AR2)

**Revision of bg-2016-238 (January 2017)**

Referee report on the revised version of the manuscript bg-2016-238
4 Dec 2016

GENERAL COMMENTS
The manuscript has been very carefully revised according to the suggestions of the both referees, and it has been significantly improved: It is now well focused, reads fluently and has all the necessary details for understanding how the different phases of the work were conducted.
I have no suggestions for corrections, despite the technical corrections listed below, and am pleased to recommend this work for publication in Biogeosciences.

**AUTHORS REPLY:** We wish to thank the reviewer for the very detailed review of our manuscript and spotting the inconsistencies, typos etc. We have corrected ALL changes as suggested (see tack changes in `bg-2016-238-text-version1.docx`). We have only inserted an AUTHOR REPLY where significant changes to the text had to be made.

TECHNICAL CORRECTIONS
Page 2, line 6: The verb is missing in '…boreal forests are known to significant stocks of organic N…'
**AUTHORS REPLY:** corrected and included the verb 'be'

Page 2, line 26: Add comma after 'forest sites'
**AUTHORS REPLY:** corrected

Page 3, lines 27-30: Complicated sentence, consider revising. Please check also the spelling of the words 'aerenchyma' and 'potential'.
**AUTHORS REPLY:** We have rephrased this sentence to:

As $CH_4$, also N2O production is a microbial process. The main drivers regulating $N_2O$ production are nitrogen, such as ammonium and nitrate, temperature and factors which regulate the ratio of aerobic to anaerobic soil microsites, such as soil moisture (Butterbach-Bahl et al., 2013). In peatlands transport through aerenchyma, is also for $N_2O$ a potential transport mechanism.

Page 4, line 25: Space missing after the full stop.
**AUTHORS REPLY:** corrected

Page 6, line 20: Comma is not needed after the parenthesis.
**AUTHORS REPLY:** corrected

Page 13, line 16: In my opinion, a comma would be a more appropriate separator between the sentences than a full stop.
**AUTHORS REPLY:** corrected

Page 14, line 8: Letter t missing in Turetsky.
**AUTHORS REPLY:** corrected

Page 15, line 4: The full word methane should be used instead of the chemical formula in the beginning of the sentence.
**AUTHORS REPLY:** corrected

Page 15, line 10: An extra full stop in the end.
**AUTHORS REPLY:** corrected

Page 15, line 19: Please check the correct quoting of Marushchak et al., 2011. The paper do not report large N2O fluxes at permafrost thaw, but from bare peat on peatlands currently elevated by permafrost, where erosion and frost action keep the surfaces bare of vegetation.
**AUTHORS REPLY**: Deleted Marushchak et al, as this reference is not appropriate

References: Please check the use of subscript/superscript letters throughout the reference list!
**AUTHORS REPLY:** corrected

Page 18, line 5 & Page 19, line 20: The links seem to be to the supplementary information, is this intentional?
**AUTHORS REPLY**: Yes, this is intentional. To clarify we have changed this by inserting 'and' in both cases, i.e.:
Geosci, 3, 617-621 and
http://www.nature.com/ngeo/journal/v3/n9/suppinfo/ngeo939_S1.html, 2010.

Page 19, line 27: A typing error in the name of the last author.
**AUTHORS REPLY:** corrected

Page 20, lines 12-14: Unnecessary capital letters.
**AUTHORS REPLY:** this reference has been removed

Page 21, lines 14-15: The formatting of the author names differs from the other references.
**AUTHORS REPLY:** corrected

Page 22, line 1: The author surnames are Tan and Zhuang, please correct the reference.
**AUTHORS REPLY:** corrected in the reference list and test p15 line 5.

**AUTHORS COMMENT**: In addition we spotted a few other small errors and have indicated changes (see the track changed version of the manuscript bg-2016-238-text-version1.docx ).